# Gas flaring activity and black carbon emissions in 2017 derived from Sentinel-3A SLSTR

Alexandre Caseiro[1,2], Berit Gehrke[1,4], Gernot Rücker[3], David Leimbach[3], and Johannes W. Kaiser[1,5]

[1]Max Planck Institute for Chemistry, Mainz, Germany
[2]Institute for Advanced Sustainability Studies, Potsdam, Germany
[3]Zebris GbR, Munich, Germany
[4]Universitymuseum, University of Bergen, Bergen, Norway
[5]Deutscher Wetterdienst, Offenbach, Germany

**Correspondence:** Alexandre Caseiro (alexandre.caseiro@mpic.de)

**Abstract.** Gas flares are a regionally and globally significant source of atmospheric pollutants. They can be detected by satellite remote sensing. We calculate the global flared gas volume and black carbon emissions in 2017 by applying (1) a previously developed hot spot detection and characterisation algorithm to all observations of the Sea and Land Surface Temperature Radiometer (SLSTR) instrument on-board the Copernicus satellite Sentinel-3A and (2) newly developed filters for identifying gas flares and corrections for calculating both flared gas volumes (Billion Cubic Meters, BCM) and black carbon (BC) emissions (g). The filter to discriminate gas flares from other hot spots uses the observed hot spot characteristics in terms of temperature and persistence. A regression function is used to correct for the variability of detection opportunities. 6232 flaring sites are identified worldwide. The best estimates of the annual flared gas volume and the BC emissions are 129 BCM with a confidence interval of [35 BCM, 419 BCM] and 73 Gg with a confidence interval of [20 Gg, 239 Gg], respectively. Comparison of our activity (*i.e.* BCM) results with those of the Visible Infrared Imaging Radiometer Suite (VIIRS) Nightfire data set and SWIR-based calculations show general agreement but distinct difference in several details. The calculation of black carbon emissions using our gas flaring data set with a newly developed dynamic assignment of emission factors lie in the range of recently published black carbon inventories, albeit towards the lower end. The data presented here can therefore be used e.g. in atmospheric dispersion simulations. The advantage of using our algorithm with Sentinel-3 data lies in the previously demonstrated ability to detect and quantify small flares, the long-term data availability from the Copernicus programme and the increased detection opportunity of global gas flare monitoring when used in conjunction with the VIIRS instruments. The flaring activity and related black carbon emissions are available as "GFlaringS3" on the Emissions of atmospheric Compounds and Compilation of Ancillary Data (ECCAD) web site (https://eccad3.sedoo.fr/#GFlaringS3, DOI 10.25326/19).

## 1 Introduction

Industrial gas flaring (GF) occurs when flammable gas is disposed of by burning, most commonly done at the tip of a stack. This can either take place as a measure for pressure relief, or to dispose of unwanted gas. In the upstream oil and gas industry

(UOG, the oil and gas industry sector that includes searching, exploring, drilling and operating the wells of crude oil and natural gas underground or underwater sources and bringing the resource to the surface) in particular, GF occurs when the associated gas can't be sold easily and is not used on-site for energy generation. (Gervet, 2007; Ismail and Umukoro, 2012; Emam, 2015)

Especially in insufficiently developed energy markets companies seem to be spared from enough economic or political incentives to collect, or convert the gas (Rahimpour and Jokar, 2012; Ojijiagwo et al., 2016). Improvements of flare gas recovery systems have been recommended for more closely monitored facilities (Zolfaghari et al., 2017; Papailias and Mavroidis, 2018).

GF negatively impacts the immediate surrounding (Akinola, 2018), for example through noise (Ismail and Umukoro, 2012; Nwoye et al., 2014), heat stress (Anomohanran, 2012; Julius, 2011) and visual pollution (Anomohanran, 2012; Ajao and Anurigwo, 2002). GF also impacts the environment on a wider scale through the emission of pollutants and greenhouse gases like carbon dioxide ($CO_2$), carbon monoxide, black carbon (BC), nitrogen oxides, polycyclic aromatic hydrocarbons, volatile organic compounds and acid rain precursors (Obioh et al., 1994; Uzomah and Sangodoyin, 2000; Nwaichi and Uzazobona, 2011; Onu et al., 2014; Li et al., 2016).

It has been estimated that between 2003 and 2012 GF produced on average 304 Tg $CO_2$ yearly, representing 0.6 % of the global anthropogenic emission of $CO_2$ equivalent (Olivier et al., 2014). Another source estimates the contribution to 400 Tg (Emam, 2015). The recovery of flare gas can play a relevant role in improving sustainability and meeting emissions targets (Ahmed Osama Abdulrahman et al., 2015; Comodi and Renzi, 2016). Elvidge et al. (2018) estimated that a suppression of GF could contribute up to 2% of the $CO_2eq$ global nationally determined contributions (NDC) under the United Nations Framework Convention on Climate Change Paris Agreement. Some countries could meet or even surpass their NDC (Yemen, Algeria and Iraq), while others could meet between one third to almost all of their NDC (Gabon, Venezuela, Iran and Sudan).

The BC emissions from GF are of particular importance because BC is a known carcinogen (Heinrich et al., 1994) as well as a short-lived climate forcer (IPCC, 2013). BC strongly affects environments such as the Arctic regions by lowering the albedo of snow-covered surfaces (Flanner et al., 2007; Stohl et al., 2013; Bond et al., 2013; Huang et al., 2015; Evangeliou et al., 2018), which has an impact on the earth's radiative balance (Doherty et al., 2010; Quinn et al., 2007; Hansen and Nazarenko, 2004) as well as adding to the Arctic amplification phenomenon (Serreze and Barry, 2011). The GF contribution to the BC global emissions was estimated to amount to 270 Gg and 210 Gg in 2005 and 2010, respectively (Klimont et al., 2017). Regionally, Stohl et al. (2013) and Cho et al. (2019) showed that GF contributes half of the near-surface BC concentration in the Arctic and explains a significant fraction of arctic warming. Fossil fuel burning was also found to be the main source of BC deposited on snow (Qi and Wang, 2019).

GF is considered an important component of atmospheric dispersion simulations (Evangeliou et al., 2018) and climate modeling (Huang and Fu, 2016) as the impact of GF emissions extends beyond immediate environmental concerns. However, information on the amount of natural gas being disposed of through stack burning and the accrued emissions is sparse. Reporting is often not enforced, inconsistent or otherwise not reliable.

Satellite remote sensing has been utilized for regional and global identification and characterization of GF (e.g. Elvidge et al., 1997; Casadio et al., 2012b, a; Anejionu et al., 2014; Faruolo et al., 2014; Chowdhury et al., 2014; Anejionu et al., 2015; Faruolo et al., 2018). The advantages and limitations of satellite remote sensing for the observation and monitoring of GF have recently

been reviewed by Obinna C. D. Anejionu (2019). The most frequently cited system is NOAA's VIIRS (National Ocean and Atmospheric Administration's Visible Infrared Imaging Radiometer Suite) Nightfire (VNF) dataset (see https://ngdc.noaa.gov/eog/viirs/download_viirs_fire.html), developed by Elvidge et al. (2013, 2016) and still under active development (Elvidge et al., 2019). VNF, which succeeded a system based on the OLS (Operational Linescan System) instrument, which has been flying on board the DMSP (Defense Meteorological Satellite Program) satellites since the 1970s (Elvidge et al., 1997, 2001) provides a consistent global survey of flaring sites and GF volumes from 2012 onwards (https://ngdc.noaa.gov/eog/viirs/download_global_flare.html).

Caseiro et al. (2018) describe an adaptation and extension of the VNF algorithm, with which observations of the Sea and Land Surface Temperature Radiometer instrument (SLSTR) on-board the Copernicus Sentinel-3 satellites can be analysed, too. The algorithm detects and quantifies hot sources, including gas flares, using the night-time readings of the shortwave infrared (SWIR), mid-infrared (MIR) and thermal infrared (TIR) channels. SLSTR observations (night-time overpasses at 10:00 PM) complement those of the VIIRS instruments (0:40 AM on JPSS-1/NOAA-20 and 1:30 AM on Suomi-NPP, our analysis predates the launch of the latter, whose measurements are not included in the present work) by filling observation gaps in the time series. Both instruments are planned to allow long term data availability.

Here, we present a new dataset of global GF sites, volumes (Billion Cubic Meters, BCM) and BC emissions (g), which is based on the observation by Sentinel-3A in 2017. Section 2 describes the newly developed methods for identifying gas flares among the observed hot sources, correcting for intermittent observations opportunities, and dynamically determining appropriate BC emission factors from the observations. To the best of our knowledge, it is the first time that the operational state of the flare, *i.e.* its temperature as determined from a satellite-based observation, is used to assign its BC emission factor. The results are presented in Section 3, including a comparison with other published methods (the VIIRS Nightfire product and the SWIR-radiance method), and the conclusions are summarised in Section 4.

## 2 Methods

### 2.1 Hot spot detection and characterization

The steps involved in the study presented here are outlined in Figure 1. We base our work on the persistent hot spots detection and characterisation algorithm described in Caseiro et al. (2018) to process a full year (2017) of Sentinel-3A's Sea and Land Surface Temperature Radiometer (SLSTR) data. A hot event at temperatures typical of a gas flare (1200 – 2500 K) will produce a local maximum in the night-time readings of the shortwave- and possibly of the mid-infrared (SWIR and MIR) channels of SLSTR. The SWIR band centered at 1.61 μm (S5) is closest to the expected spectral radiance maximum and serves as the primary detection band. Hot SWIR and MIR pixels are searched for using a contextual approach and their radiances are extracted. The radiometrically calibrated radiances of the different bands are used to fit the sum of two Planck curves: one representing the hot source and the other the cool background. Both curves are fitted to the night-time spectral infrared (short-wave, mid-wave and thermal infrared: SWIR, MIR and TIR) observations to characterize temperature (T) and area (A) of the observed gas flares. Detections are divided into spurious signals (if the temperature retrieval is outside the 500–5000 K range)

and valid hot spots. Hot spots are further divided into high accuracy (detection in at least another band besides S5, i.e. S6 or S7 or F1, meaning that with the LWIR there are at least 4 points for the Planck curve fit, and at least 3 cloud-free pixels in the background to compute the background radiances) and low-accuracy. Subsequently, the radiative power (RP) of the flare is calculated with the Stefan-Boltzmann equation.

While in principle the methodology used is based on VNF developed for Visible Infrared Imaging Radiometer Suite (VIIRS) (Elvidge et al., 2013, 2016), it differs by (1) analysing the radiances of clusters of contiguous hot pixels instead of treating spatial maxima as individual pixels and (2) using the TIR channels when fitting the sum of the two Planck curves. By considering all the hot pixels and therefore using all the information on the spatial scale we expect our representation to be more realistic for the characterisation of large gas flare arrays. The use of the TIR channels implies a more stringent constraining of the
background temperature. These options may lead to the retrieval of a lower flaring temperatures than those retrieved using the VNF approach (Caseiro et al., 2018). The lack of ground truth hinders the evaluation of the accuracy of the different options.

    Caseiro et al. (2018) already tested the method within a limited time-span using oil and/or gas producing regions and compared the results to the VNF "flares only" product. The results showed good agreement of the hot source detection when investigating persistent hot spots with the advantage of the Sentinel-3A's SLSTR algorithm in detecting and quantifying smaller
flares, due to the night-time availability of a second SWIR channel. The characterisation in terms of temperature, area and radiative power reached similar values. However, temperatures were slightly lower, while areas and radiative power were slightly larger, in the SLSTR-based results. Note that VIIRS Nightfire also uses two SWIR channels since January 2018 and that the detections are now conducted by two VIIRS instruments (at 0:40 and 1:30 AM).

## 2.2   Hot spot classification

### 2.2.1   Volcano filter

As discussed in (Caseiro et al., 2018), the criteria adopted to filter GFs are not enough to avoid false detections due to volcanoes, suggesting the use of masks. In order to filter out volcanic activity from, we use the data available from the Global Volcanic Program of the Smithsonian institution (Venzke, 2013). A mask at 0.025° resolution was calculated from all eruptions in 2017. It also contains buffering zones stretching to 0.25° distance from the volcanic center coordinates because these coordinates
may not correspond to the location of the thermal radiation emission: Many volcanoes do not consist of a single edifice and many individual eruptive fissures through which lava erupts may be present in a volcanic field (Siebert et al., 2010).

### 2.2.2   Discrimination of gas flares from other industrial hot sources

In order to discriminate gas flares from other night-time hot spots we investigate persistence and retrieval temperature filtering.

    We project all the hot spot detections onto a $0.025 \times 0.025$ degrees global grid. This is a spatial resolution that reflects the
reality on the ground, where several flares, separated by tens to hundreds of meters, may be operated within a single oil and gas facility, while facilities are tens of kilometers apart. This resolution is also adequate to the pixel footprint of medium-resolution imagers (Facchinelli et al., 2019). The following quantities are then computed for each grid cell: the number of hot spot

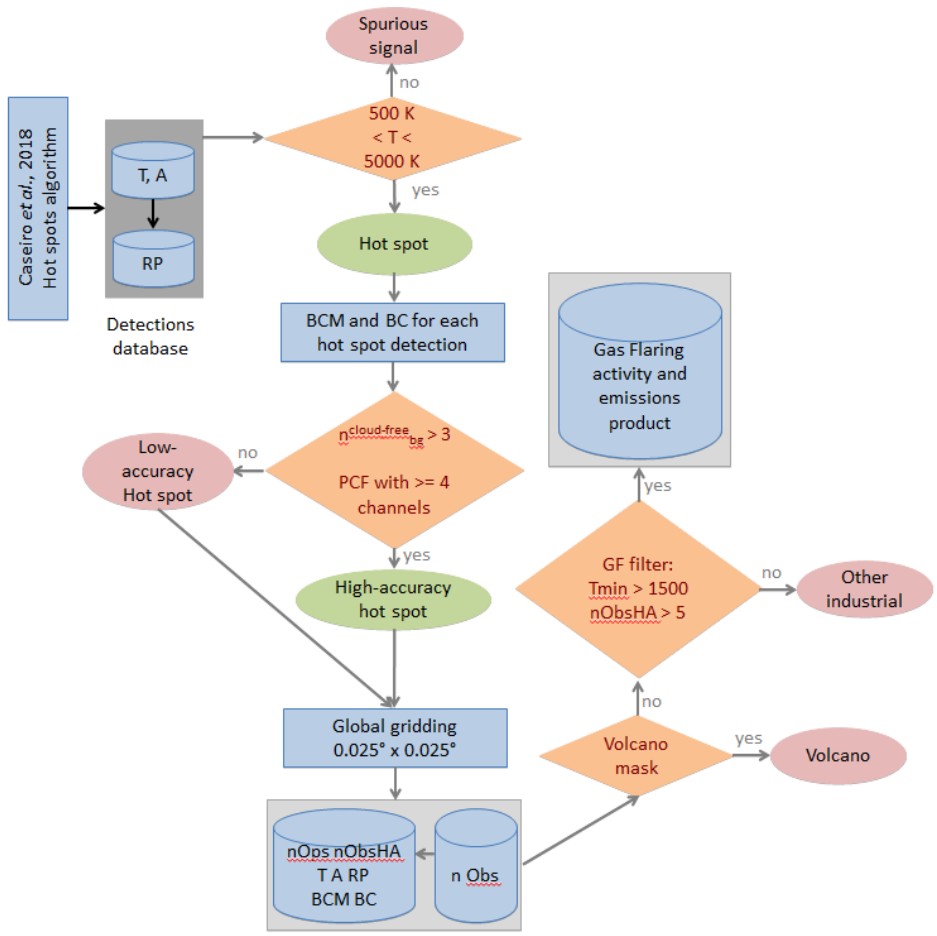

**Figure 1.** Flowchart of the methodology used in the present work. Starting point is the hot spot algorithm detection described in Caseiro et al. (2018) which, after applying a filter on the retrieved temperature (500 K < T < 5000 K), produces a database of valid hot spots detections. For each of these valid hot spot detections, the activity (Billion Cubic Meters, BCM) and Black Carbon (BC) emissions are computed. The valid hot spot detections are further filtered: detection in at least another band besides S5 (S6 or S7 or F1, meaning that with the LWIR there are at least 4 points for the Planck curve fit, PCF) and at least 3 cloud-free pixels in the background to compute the background radiances. This step produces a database of high-accuracy hot spots detections. All hot spot detections are gridded into a global $0.025° \times 0.025°$ grid. For each grid cell, the following quantities are computed: the number of valid detections ($n_{Obs}$), of high-accuracy detections ($n_{Obs_{HA}}$) and of days of operation ($n_{Ops}$) (see below, Section 2.2.2), temperature, area and radiative power (minimum, average, maximum and standard deviation), the date of the first and the last detection and the sum of the activity (BCM) and BC emissions. A volcano filter is applied to the global grid in order to mask out volcanic activity (see Section 2.2.1). The last step is the discrimination of grid cells for flaring activity (see Section 2.2.2).

detections ($n_{Obs}$), the number of high-accuracy (Caseiro et al., 2018) hot spot detections ($n_{ObsHA}$), the minimum, maximum, average and standard deviation of the high-accuracy hot spot retrieved temperature ($T$), area ($A$) and radiative power ($RP$) as well as the earliest and the latest hot spot observation date.

The temperature value used in the selection process of the discrimination strategy is based on Elvidge et al. (2016) and on the recent work by Liu et al. (2018), who derived gas flaring temperatures of 1000 K to 2600 K from the VIIRS Nightfire database, depending on the type of operation (shale oil or gas, offshore, onshore or refinery). According to these authors, most of the gas flares display temperatures between 1650 K and 1850 K. However, temperatures can occasionally be as low as 1300 K. We therefore test for the minimum and/or for the maximum temperature for all the high-accuracy detections within a grid cell ($T_{min}$ and $T_{max}$, respectively). The temperature range reported by Elvidge et al. (2016) and Liu et al. (2018) (1650 – 1850 K) overlaps with particularly hot detections from the coal chemical industry and steel plants. Therefore, additional criteria are needed for identifying gas flares in the hot source dataset, which are explored in the following paragraphs.

In order to select the discriminating strategy we test several subsets of the gridded high-accuracy hot spot database. For each of the 8 subsets described in Table 1, a sample of 100 random onshore grid cells complying to the defined thresholds have been tested by examining high-resolution imagery (Google Earth) and the locations are classified into four categories:

**Flare** ($F$) flaring site with visible flame

**Likely** ($L$) industrial or oil extraction site with typical flaring infrastructure but no visible flame

**Unlikely** ($U$) industrial site without typical flaring infrastructure

**No industry** ($N$) e.g. agricultural or forested area.

The subsets are characterized in terms of user's accuracy (UA, Equation 1) and commission error (C, Equation 2), where F, L, N, U denote the number of grid cells in the corresponding category:

$$User's\ accuracy = \left(F + \frac{L}{2}\right) \pm \frac{L}{2} \tag{1}$$

$$Commission\ error = \left(N + \frac{U}{2}\right) \pm \frac{U}{2} \tag{2}$$

The user's accuracy is the accuracy from the point of view of the user. This metric represents the frequency with which a class on the map corresponds to the ground truth. In our case, UA lies between at least the verified flaring locations and at most the sum of the verified and the likely flaring locations. The commission error is calculated by reviewing the classified sites for incorrect classifications. In the present work, the commission error lies between at least the locations without any industry or infrastructure and at most the sum of the locations without any industry or infrastructure and the unlikely flaring locations.

In some industrial areas, facilities that use gas flares may be close to other hot spots, such as iron smelters or steel mills. In those cases, a commission error occurs when the flare is off and our methodology samples the other hot spot. The omission

**Table 1.** User's accuracy (UA, %) and commission error (C, %) of the hot spot discrimination strategies considered. $n_{Obs}$ is the number of hot spot detections within a grid cell, $n_{ObsHA}$ is the number of high-accuracy hot spot detections within a grid cell, $T_{min}$ is the minimum temperature retrieved among all the hot spots detected within a grid cell, $T_{max}$ is the maximum temperature retrieved among all the hot spots detected within a grid cell, $n_{cells}$ is the number of grid cells that comply with the thresholds. We have tried 8 combinations (discrimination strategies) of thresholds on those variables. Each column represents a tested discrimination strategy.

| combination | #1 | #2 | #3 | #4 | #5 | #6 | #7 | #8 |
|---|---|---|---|---|---|---|---|---|
| $n_{Obs} >$ | – | 3 | 4 | – | – | – | – | – |
| $n_{ObsHA} >$ | 2 | 2 | 2 | 5 | 5 | 5 | 7 | 7 |
| $T_{min}\ (K) >$ | 1000 | 1000 | – | – | – | – | – | – |
| $T_{max}\ (K) >$ | 1400 | 1400 | 800 | 1200 | 1500 | 1800 | 1200 | 1500 |
| $n_{cells}$ | 6733 | 5872 | 9469 | 6817 | 6232 | 5485 | 5527 | 5129 |
| UA | 84±6 | 86±8 | 60±10 | 77±13 | 85±11 | 88±10 | 73±14 | 87±11 |
| C | 7±3 | 4±2 | 19±11 | 6±4 | 3±1 | 1±1 | 8±5 | 2±1 |

error can be divided into two categories: flares that the hot spot detection and characterization algorithm failed to detect and flares that were detected as hot spots but which the discrimination strategy left out. The former will be the same for any discrimination procedure considered.

In order to discriminate gas flares from other hot spots we discriminate hot spots based on their persistency ($n_{Obs}$, number of hot spot detections, and $n_{ObsHA}$, number of high-accuracy hot spot detections) and on their temperature time series ($T_{min}$ and $T_{max}$, the minimum and the maximum temperature retrieved among all the hot spots detected within a grid cell). We have tried 8 combinations (discrimination strategies) of thresholds on those variables (Table 1). For each of the 8 combinations, we examine high-resolution imagery for 100 random onshore locations (800 in total) in order to verify the presence of a gas flare. The goal is to maximize user's accuracy (UA) and minimize commission error (C) while minimizing the omission error (here, the variation in $n_{cells}$ is used as a proxy). The discrimination strategy #5 (Table 1) was selected as the most suitable: detections located in grid cells where the yearly maximum temperature retrieval is above 1500 K and a persistency and quality criterion ($n_{ObsHA} > 5$) is met are considered as originating from gas flares and called flaring locations hereafter. This results in a relatively large number of grid cells with detections, i.e. low omission error, combined with a high user's accuracy and a low commission error of 85±11% and 3±1%, according to Equations 1 and 2, respectively. Inserting a limitation on the minimum temperature or increasing the maximum temperature or the number of high-accuracy observations would not significantly increase the user's accuracy or decrease the commission error, but would reduce the number of grid cells complying with the discrimination criteria by several hundreds, which we interpret as an increase in the omission error. Although our discrimination strategy is satisfactory at the planetary level, global aggregating approaches may miss individual occurrences, in our case especially where the oil exploitation frontier is under expansion (Facchinelli et al., 2019).

## 2.3 Determination of flared volumes and black carbon emissions

In order to compute the activity and emissions, we estimated the number of days of operation per site by expressing the maximum number of observations as a function of latitude (in $10°$ bins, see Figure 2). The function computes the maximum number of observations $n_{Obs_{max}}$ per grid cell for a given latitude which we assume expresses a continuous hot spot (365 days a year). The number of days of operation $n_{Ops}$ is then estimated by scaling following Equation 3. By this scaling, we approximately correct for the limitations of gas flaring detection from space (cloud cover and overpass frequency).

$$n_{Ops} = n_{Obs} \times \frac{365}{n_{Obs_{max}}} \qquad (3)$$

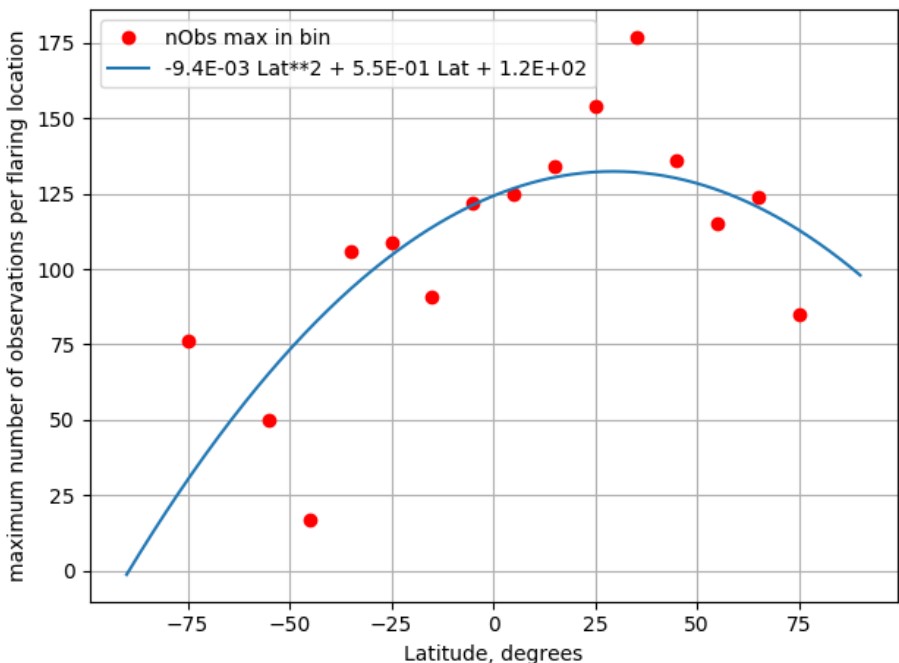

**Figure 2.** Maximum number of observations per grid cell as a function of Latitude. The latitude values are binned ($10°$). The function expresses the maximum number of observations $n_{Obs_{max}}$, which we assume as a continuous operation of the hot source, as a function of latitude.

For the estimation of the flared volume, we applied the calibration derived by Elvidge et al. (2016) to each single detection within a flaring location grid cell. The calibration relationship uses a modified formulation of the radiative power to output the yearly flared volumes (in Billion Cubic Meters, BCM), which is then scaled to a daily activity in m$^3$.

The lower bound $BCM_{min}$ for the activity estimate assumes that the flare is only active on the $n_{Obs}$ days with hot spot detections. The upper bound for the activity estimate assumes that the flare is constantly active:

$$BCM_{max} = BCM_{min} \times \frac{365}{n_{Obs}} \tag{4}$$

The best activity estimate corresponds to assuming that the flare is active for $n_{Ops}$ days:

$$BCM_{best} = BCM_{min} \times \frac{n_{Ops}}{n_{Obs}} \tag{5}$$

The emission of black carbon (BC) from gas flares are estimated using reported emission factors (EFs), using the traditional approach that the emission equals the multiplication of the activity (*i.e.* BCM) by the EF. Klimont et al. (2017) recognized the limited number of measurements of flaring emissions. Here, we maximize the use of the limited information available on the EFs by combining it with the individual observed flare characteristics.

EF data are scarce and the few published results span a considerable range of values, see McEwen and Johnson (2012); Stohl et al. (2013); Schwarz et al. (2015); Huang and Fu (2016); Weyant et al. (2016); Klimont et al. (2017). In the present work, we apply the same concept of a varying EF as the one used by Klimont et al. (2017), who used the country of origin as an indication of the flare operation. However, instead of the physical location of the flare, we use the retrieved flaring temperature as an indication of the combustion completeness. Flaring temperatures close to the adiabatic flame temperature for natural gas (around 2500 K) are associated with more complete combustion and therefore lower BC emissions ($0.5\,g.m^{-3}$). On the other hand, low flaring temperatures (700 K and below) are associated with higher BC emissions ($1.75\,g.m^{-3}$). Between the two extremes, provided by the work of Klimont et al. (2017), the BC emission is scaled linearly as a function of the flaring temperature (see Figure 3). To the best of our knowledge, this is the first time that operating conditions are taken into consideration when assigning the EF, and that the operating conditions are derived from direct observation.

The country-level BCM and BC estimates are computed by summing the individual flares estimates within the borders of each country and its exclusive economic zone.

With this methodology we estimate a wide range of possible activity (BCM) and BC emissions (g), where our best estimate falls between the 'flaring only when there is a detection' ($BCM_{min}$ and $BC_{min}$) and 'constant flaring' ($BCM_{max}$ and $BC_{max}$). We consider the range between $BCM_{min}$ and $BCM_{max}$ to be the confidence interval for the possible activity of a flaring site. Similarly, $[BC_{min}, BC_{max}]$ is the confidence interval for the possible emission.

## 3 Results

### 3.1 Flaring locations

All the hot spots detected globally for 2017 using the algorithm as described in Figure 1 are shown in Figure 4. Their classification is shown in Figure 5

Noise, such as detections in the open ocean and the South Atlantic anomaly, is filtered out when considering only the high-accuracy hot spots as defined by Caseiro et al. (2018). There are 427202 high-accuracy hot spots in the dataset. Their

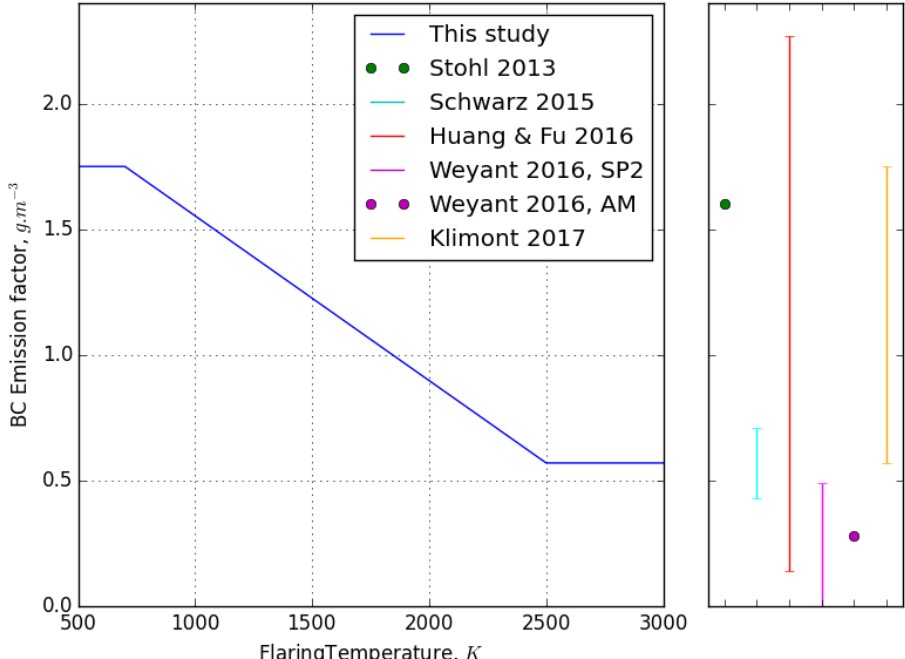

**Figure 3.** Left: BC emission factor function used in the present study. The emission factor is bounded by the extremes of the range used in the GAINS model (Klimont et al., 2017) and scaled as a function of the flaring temperature, used as an indication of combustion completeness. Right: For comparison purposes, the EFs derived by Stohl et al. (2013); Schwarz et al. (2015); Huang and Fu (2016), Weyant et al. (2016) (using the Single Particle Soot Photometer, SP2, or the Particle Soot Absorption Photometer, AM) and Klimont et al. (2017).

distribution pattern suggests that they encompass a large range of hot sources, such as wildfires, volcanoes and industrial sources.

After gridding, filtering out active volcanoes (71 volcanic eruptions in 2017) and retaining the grid cells passing the GF criteria, the number of globally detected flaring sites in 2017 is 6232. part of paragraph removed.

5   Russia (985) and the United States (917) are the countries with the highest number of flaring locations, see Figure 6. The third country is Iran (441) with less than half of the top two countries. The other countries with more than 300 flaring locations are China (365) and Algeria (324). These five countries account for about one half of the global flaring locations.

part of paragraph removed.

## 3.2   Flaring characteristics

Statistics on retrieved temperature are reported in Figures 7 and 8.

10   The average temperature at the flaring locations approximately ranges from 950 K to 2250 K. This is slightly smaller than the range 750–2500 reported by Liu et al. (2018), who used VIIRS Nightfire data. This conforms to the findings of Caseiro

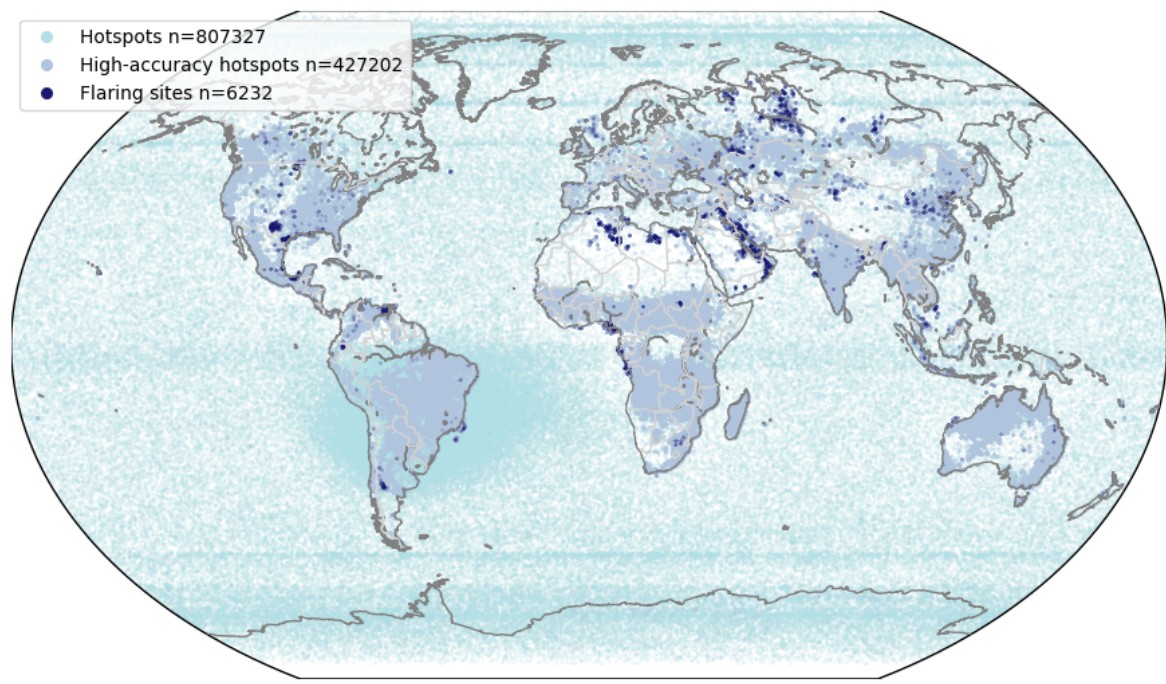

**Figure 4.** Location of all the 807327 hot spots (all detections database filtered for spurious signals, see Figure 1), of all the 427202 high-accuracy hot spots (hot spots filtered for cloudiness and accuracy of the Planck curve fitting, see Figure 1) and of all the 6232 flaring locations (grid cells with at least 5 high-accuracy hot spots and a maximum retrieved temperature above 1500 K), detected in 2017.

et al. (2018). The distribution of average retrieved temperatures confirms the bi-modal distribution with modes around 1750 K and 1200 K that has also been observed by VIIRS (Elvidge et al., 2013).

Statistics on retrieved radiative power are reported in Figures 9 to 10.

The average radiative power at flaring locations range from a few tens of MW up to approximately 1 GW (Figure 9), spanning the 5 orders of magnitude reported by Elvidge et al. (2016). Fisher and Wooster (2019) report a global FRP spanning from a few tens up to approximately one hundred MW, with a median between 1 and 2 MW. Our results are very similar.

There were 197610 detections at flaring locations in the SLSTR dataset, most were clusters of up to 20 S5 pixels, as shown in Figure 11.

2013 in the final manuscript

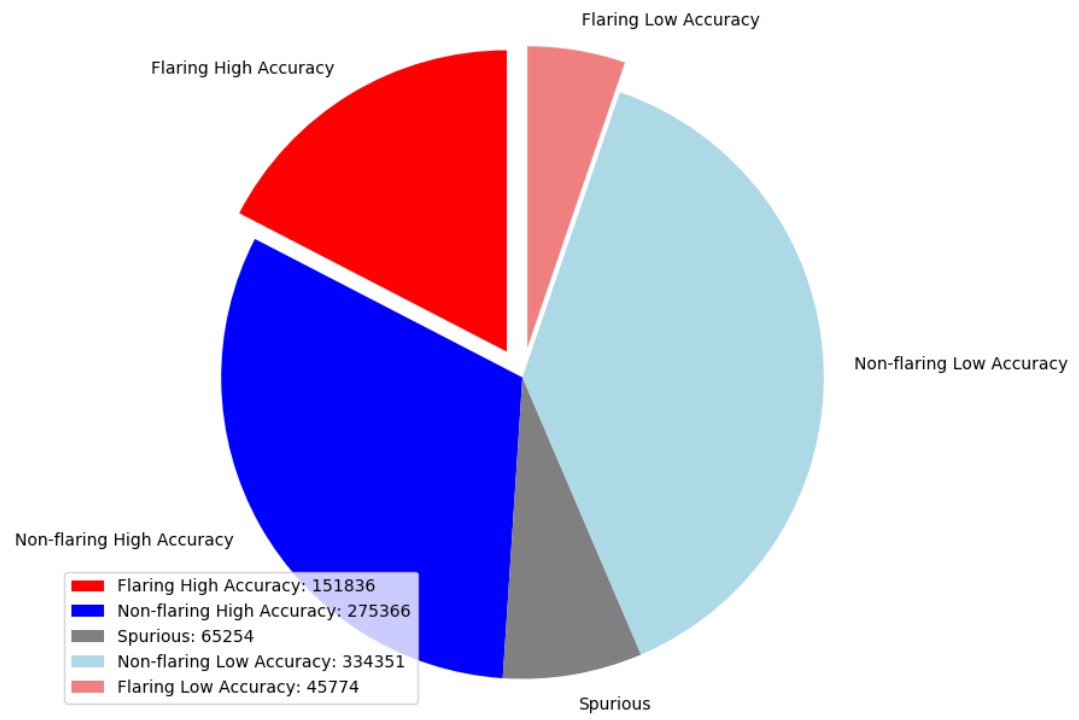

**Figure 5.** Classification of the 872581 detections: 65254 are spurious signals, 807327 are hotspots. Of the 807327 hotspots, 427202 are high-accuracy (filtered for cloudiness and accuracy of the Planck curve fitting, see Figure 1) and 380125 are low accuracy. After gridding $(0.025° \times 0.025°$ global grid), 6232 grid cells are classified as flaring locations (grid cells with at least 5 high-accuracy hot spots and a maximum retrieved temperature above 1500 K). Flaring locations comprise 197610 hotspots, of which 151836 are high accuracy and 45774 are low accuracy.

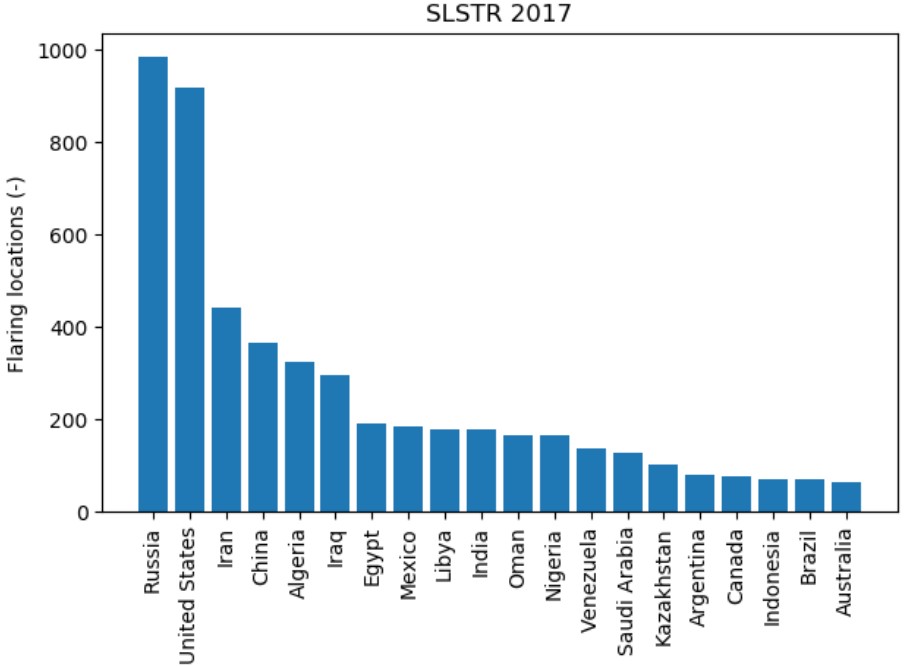

**Figure 6.** Number of flaring locations by country, top 20.

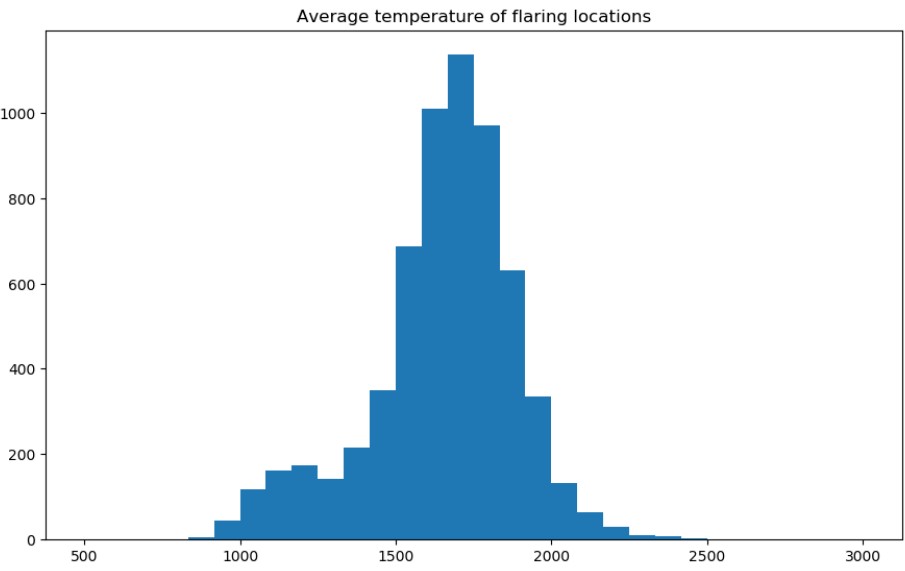

**Figure 7.** Distribution of the average retrieved temperatures (K) for the flaring locations.

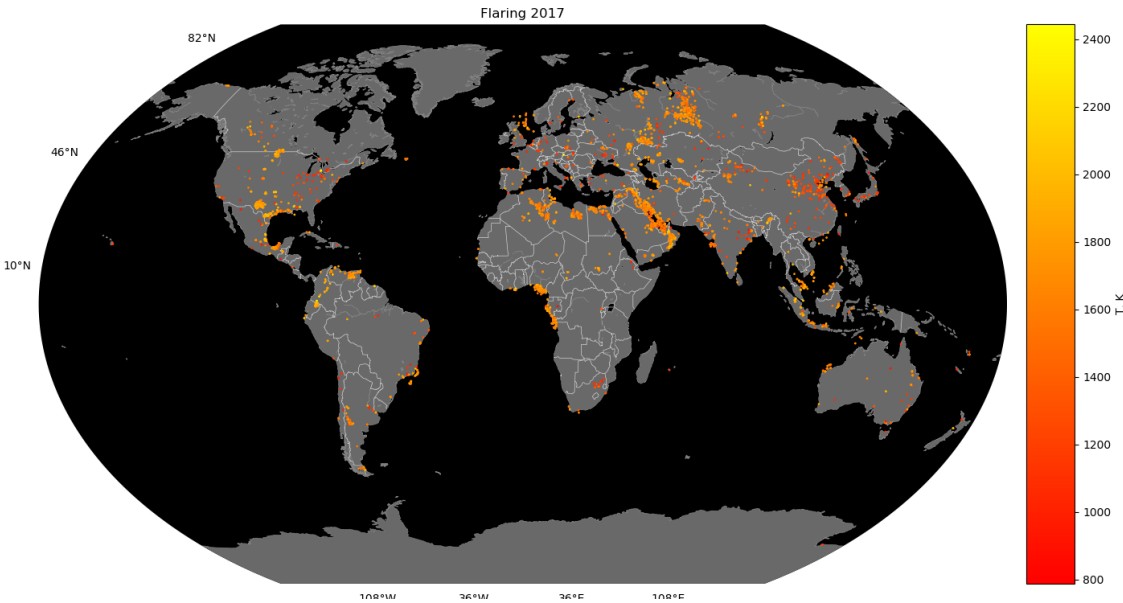

**Figure 8.** Average flaring temperature (K) at the 6232 flaring locations.

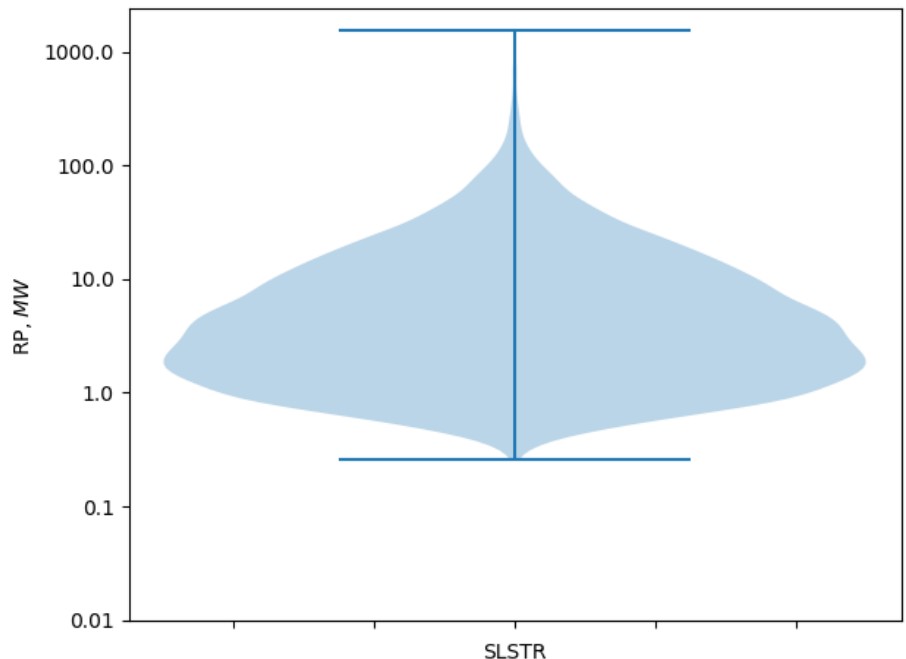

**Figure 9.** Distribution of the average radiative power (MW) for the 6232 global flaring locations.

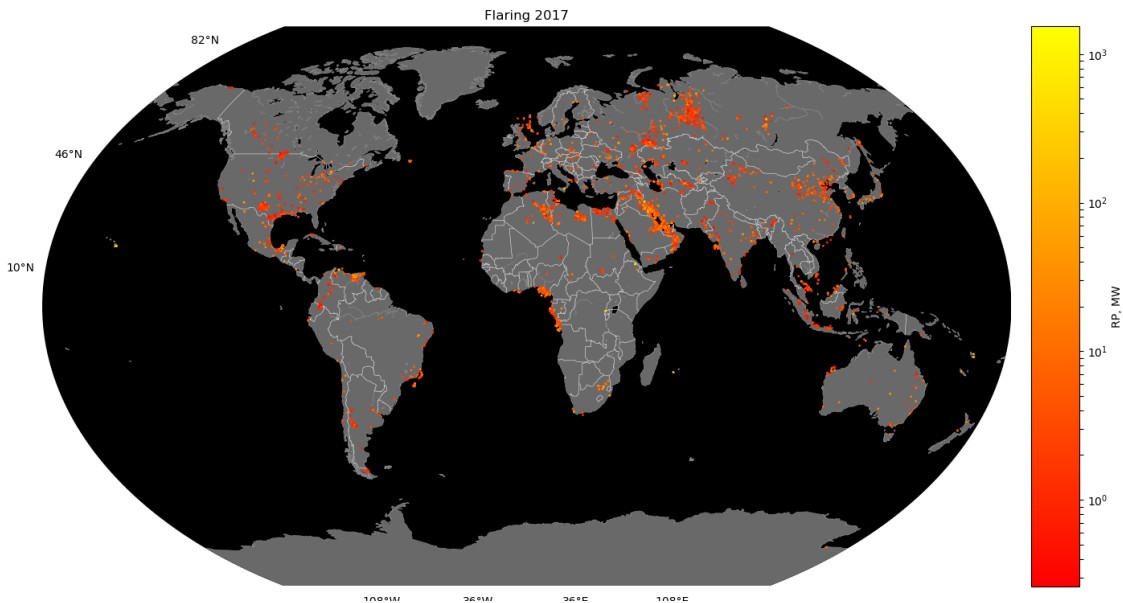

**Figure 10.** Average radiative power (MW) at the 6232 flaring locations.

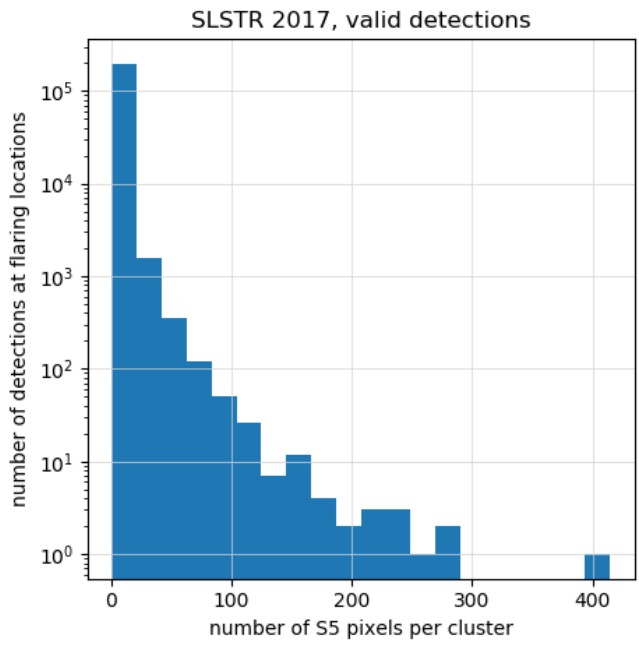

**Figure 11.** Distribution of the number of S5 pixels per cluster for the 197610 valid detections at the 6232 flaring locations. Note the log scale on the y-axis.

### 3.3 Flared volumes

The activity for all the individual flaring locations is shown in Figure 12. The most active flare burned 0.623 BCM in 2017 and is located south of Punta de Mata in Venezuela. This is also where Elvidge et al. (2016) found the most active flare for 2012, though more active (1.13 BCM).

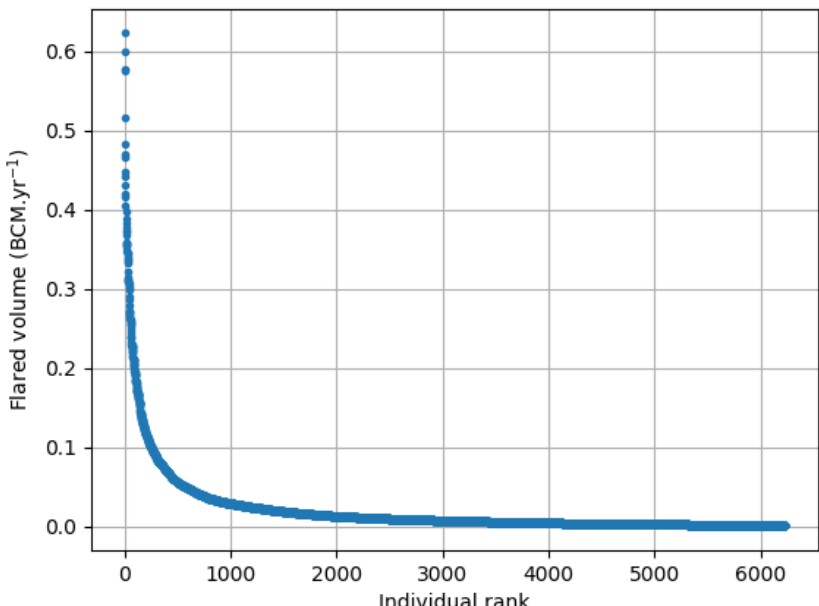

**Figure 12.** Flaring activity (best estimate, BCM per year) for each individual flaring location (6232 in total).

5    Our best estimate for the global flared volume for 2017 is 129 BCM with a confidence interval of [35,419] BCM.
Approximately half of the global flared volume originates from 4 countries only: Iraq (19 BCM), Iran (18 BCM), Russia (18 BCM) and Algeria (11 BCM) (Figure 13).

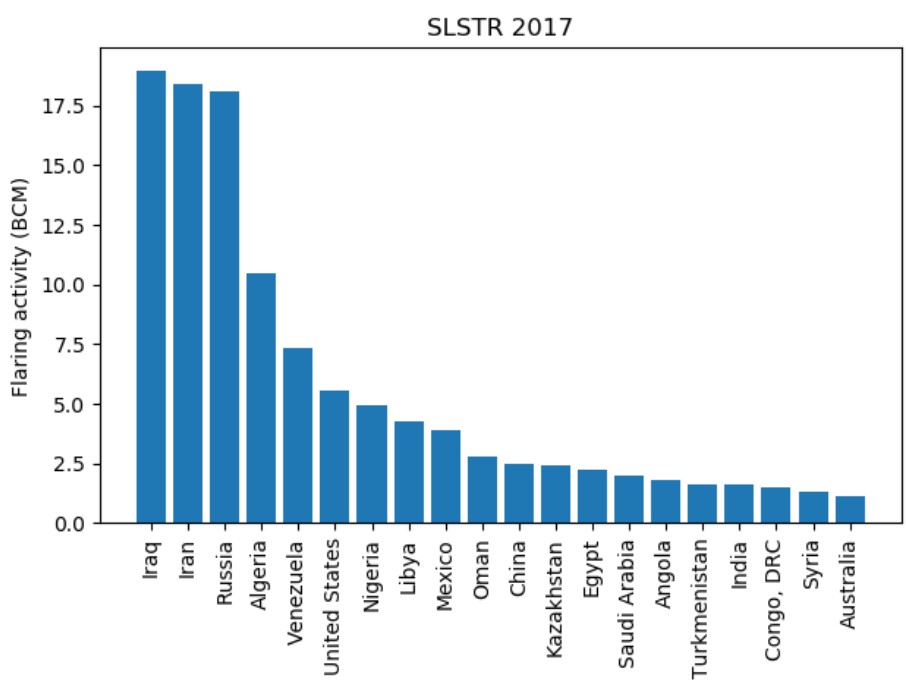

**Figure 13.** Flaring activity (best estimate, BCM per year) by country, top 20.

### 3.4 BC emissions

The black carbon (BC) emissions for all the individual flaring locations are shown in Figure 14. The most active flare emitted 0.35 Gg BC in 2017 (south of Punta de Mata).

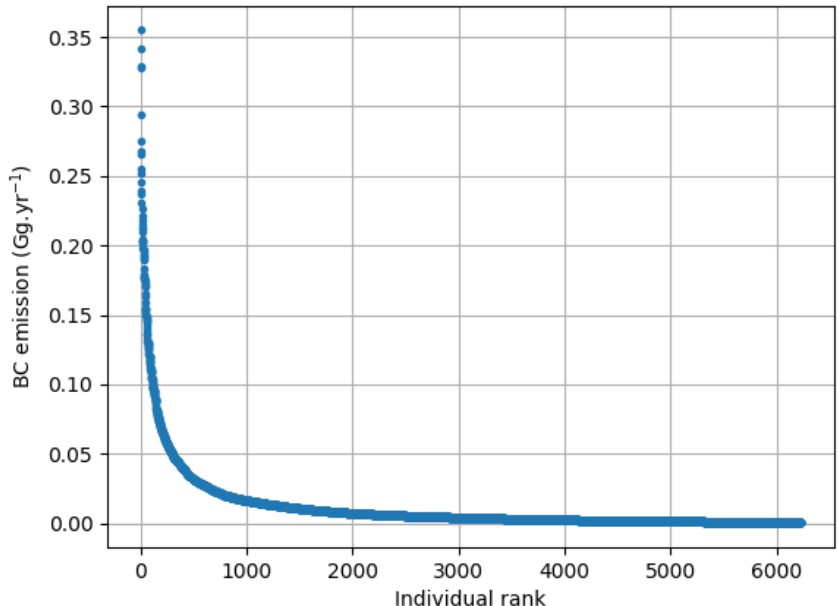

**Figure 14.** Flaring BC emissions (best estimate, Gg per year) for each individual flaring location (6232 in total).

Our best estimate for the global BC emissions from gas flaring is 73 Gg (lower estimate: 20 Gg, upper estimate: 239 Gg). As for the flared volume, 4 countries (see Figure 15) account for more than half of that value: Iraq (11 Gg), Iran (11 Gg), Russia (10 Gg) and Algeria (6.0 Gg).

The global value determined here (73 Gg) is about one third of the estimate from the Greenhouse Gas – Air Pollution Interactions and Synergies (GAINS) model (ECLIPSE V5a global emission fields) (Klimont et al., 2017): 270 Gg and 210 Gg in 2005 and 2010, respectively. Such a decrease between 2010 and 2017 may be unlikely, given that the production within the UOG did not decrease in such a significant way. Rather than a decrease in the activity, or a shift in the emission factors (EF), the discrepancy might be due to how the EF is determined (see Section 2.3). Indeed, Klimont et al. (2017) used satellite-based retrievals for the flaring locations and their characteristics (VNF). Since the VNF methodology for the detection and characterisation of flares has several similarities with the one presented here, the difference in the computed emissions are more likely to come from a difference in the EF used. Therefore we consider that part of the equation as the most relevant in interpreting the difference between our results and the results from Klimont et al. (2017) . Assigning the EF at the operational level, for each detection, seems more realistic than assigning it at the country level.

Our global estimate is larger than the global extrapolation made from a flaring emission study in the Bakken field by Weyant et al. (2016): 20±6 Gg. The dynamic assignment of the EF, based on the single temperature retrieval, should be closer to reality than globally extrapolating the EF measured at a single field. Huang and Fu (2016) used the VIIRS Nightfire dataset for the flaring activity and emission factors derived from the flared gas chemical composition to compute the BC emissions. Their

5   estimates range from approximately 140 and 200 Gg per year between 1994 and 2012. VIIRS Nightfire was also the basis for the determination of atmospheric emissions from gas flaring by Doumbia et al. (2019) for the African continent. The authors computed a BC emission in 2005 between 6.2 Gg and 141 Gg. For this continent, our results yield 16 Gg.

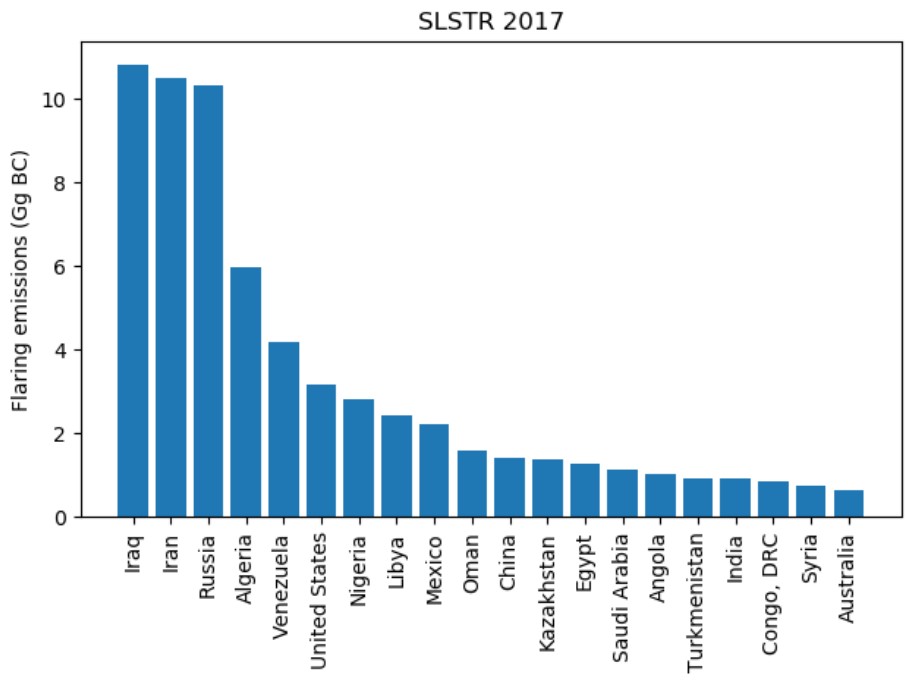

**Figure 15.** Flaring BC emissions (best estimate, Gg per year) by country, top 20.

### 3.5 Comparison with VIIRS Nightfire

Visible Infrared Imaging Radiometer Suite (VIIRS) Nightfire (VNF) 2017 flaring results are being made available by the National Geophysical Data Center of the National Oceanic and Atmospheric Administration of the United States at https://www.ngdc.noaa.gov/eog/viirs/download_global_flare.html. In this section, we perform an in-depth comparison of our results with those from VNF in terms of flaring locations, temperature and activity.

Figure 16 shows all Nightfire flaring detections in 2017 projected onto the same $0.025° \times 0.025°$ global grid as used for our results. The original VNF database lists a total of 10825 observations. After the projection, these produced 10185 flaring locations. This is an indication that the gridding spatial resolution used is adequate.

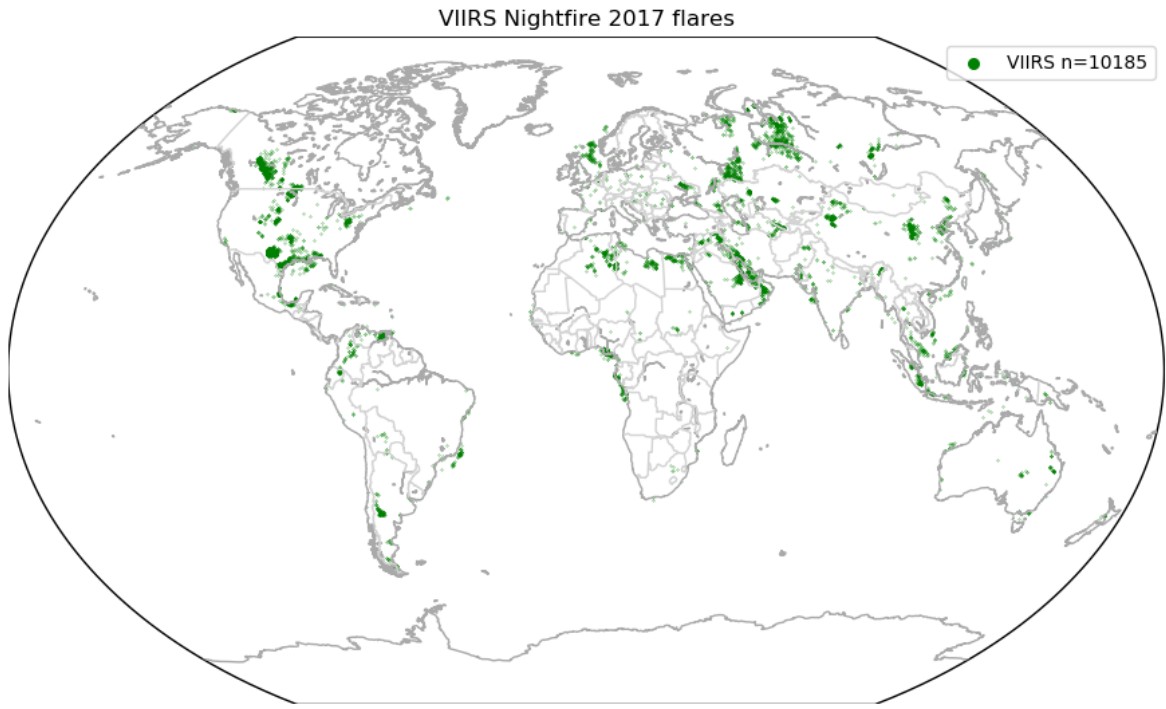

**Figure 16.** VIIRS Nightfire flaring locations for 2017. We considered every record in the VIIRS Nightfire data as a flare, i.e. "upstream" (10009), "downstream oil" (718) and "downstream gas" (101). After gridding, 10185 cells were populated and are considered as flaring sites. A detailed spatial analysis between these locations and those found by our methodology is shown in Figure 18

A comparison between Figures 4 and 16 reveals a general global agreement between the methods in terms of detections, with the exception of the regions of tar sands in Canada and shale gas in the US. Gas flaring regions are more populated by the VIIRS-based methodology than by the present work. Indeed, we find 6232 flaring sites against 10185 from the gridded VIIRS Nightfire data. The discrepancy might arise from different gas flare identification criteria: For SLSTR, a flaring site is a grid

cell that has a minimum retrieved flaring temperature >1500K and at least 5 high-accuracy detections throughout the year. In the VNF dataset, a flaring site is assigned to any grid cell for which the dataset has at least one record.

Among the 10185 VNF flaring locations and the 6232 flaring locations reported in the present work, 2964 are coincident. Figure 17 compares the average retrieved flaring temperature and the total BCM flared at these coincident flaring locations. Both methodologies retrieve temperatures roughly in the same range, with the points centered around a unique region and an observable trend in the comparison. Due to these characteristics and taking into consideration that a satellite-based observation is nothing but the snapshot of a moment in time, in this particular case, the snapshot of an inconstant phenomenon, we find the correlation reasonably good. The SLSTR-based method retrieved temperature are more spread out over the values' range than in the VNF product though. The BCM retrievals are well correlated with $R^2 = 0.74$. This shows that the differences, and presumably errors, in the temperature and flare area retrievals partly compensate each other. The regression line shows that the BCM in the VNF dataset is on average 16% larger than the one retrieved by the SLSTR-based method.

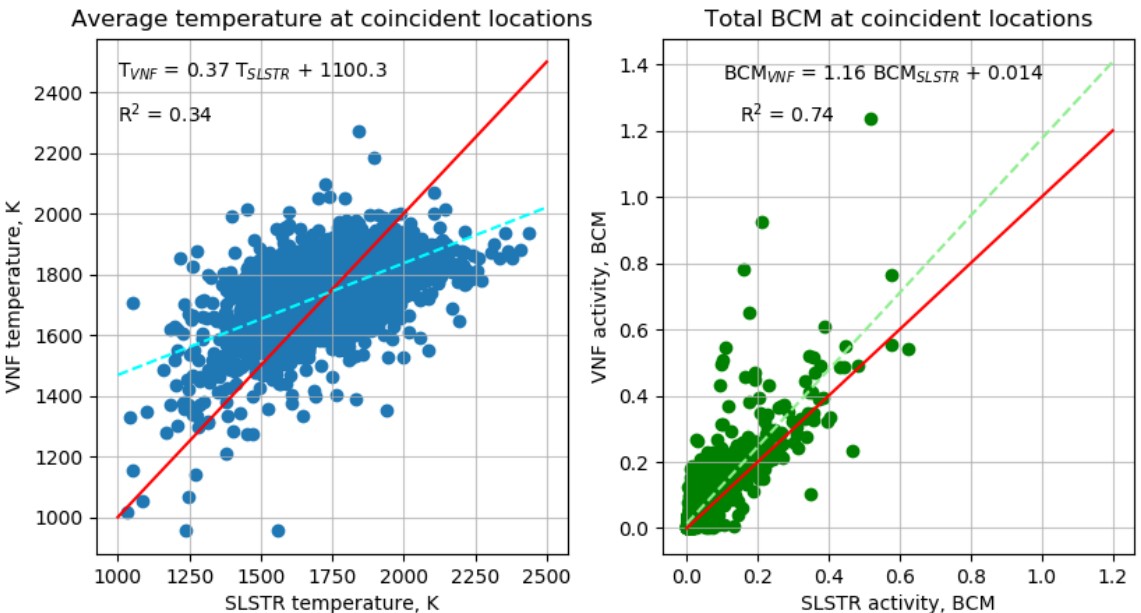

**Figure 17.** Comparison of the average temperature and the total BCM retrieved by VIIRS Nightfire and the SLSTR-based method at the 2964 coincident flaring locations. The red line represents the 1:1 relationship. The dashed line represents the linear regression between both methodologies.

7221 (70%) VNF flaring locations and 3268 (52%) SLSTR-based flaring locations are not coincident. The minimum distance from a VNF flaring location and the closest SLSTR-based flaring location, and vice-versa, are shown in Figure 18.

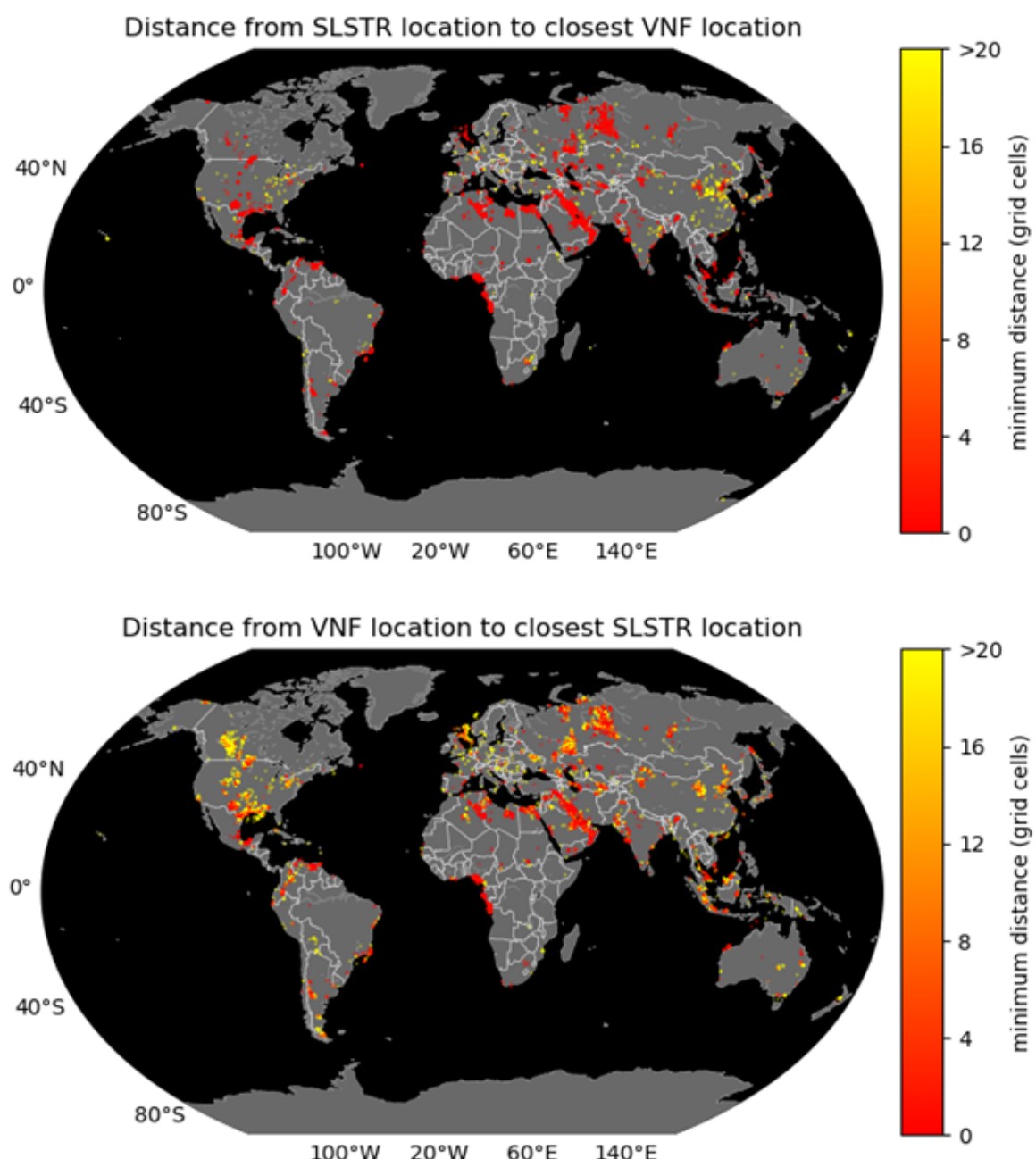

**Figure 18.** Minimum distance from a SLSTR flaring locations at the closest VNF flaring location (upper map). Minimum distance from a VNF flaring locations at the closest SLSTR flaring location (lower map). The distances are measured in grid cells (0.025° × 0.025° global grid). Of the 6232 VNF flaring sites, 2964 are coincident with SLSTR flaring sites and 2562 are directly adjacent. Of the 10185 VNF flaring sites, 2964 are coincident with SLSTR flaring sites and 1507 are directly adjacent.

**Table 2.** Comparison between the number ($n$) of flaring locations by VNF and the SLSTR-based method: total number, coincident, adjacent ($<2$ grid cells away), clustering adjacent ($\geq 2$ and $\leq 4$ grid cells away) and distant ($\leq 4$ grid cells away) locations. The sum of the activity (in BCM) for each is also shown.

| | | total flaring locations | coincident locations | adjacent locations | clustering adjacent locations | distant locations |
|---|---|---|---|---|---|---|
| distance $d$ | grid cells | | $d=0$ | $0<d<2$ | $2\leq d\leq 4$ | $d>4$ |
| SLSTR-based | $n$ | 6232 | 2964 | 2562 | 73 | 633 |
| | BCM | 129 | 78 | 39 | 0.9 | 10 |
| VNF | $n$ | 10185 | 2964 | 1507 | 1651 | 4063 |
| | BCM | 151 | 132 | 7.3 | 3.1 | 8.7 |

We interpret the large difference in the number of detections as the effect of several factors: (1) small geolocation errors, (2) clustering versus local maxima detections, (3) difference in the observation opportunities. In Table 2 we attempt to quantify the importance of these factors.

Distances up to one grid cells (adjacent cells) may arise from small geolocation inaccuracies. This accounts for 2562 SLSTR-
based locations and for 1507 VNF locations.

The Nightfire algorithm considers pixels which are local maxima and our methodology aggregates contiguous hot pixels within a unique cluster. These clusters may be very large, contain more than one local maximum and spread over grid cells, but only one cell is finally considered as its location. This was already indicated in Caseiro et al. (2018), where data retrieved from a natural gas and condensates production site in the Yamal peninsula (northern Siberia) was compared between VIIRS,
SLSTR and HSRS (Hot Spot Recognition System). In that particular case, the methodologies relying on clustering of hot pixels (SLSTR and HSRS) reported fewer gas flaring sites than VIIRS Nightfire. In the current database, about half of the 197610 hot spots at flaring locations were clusters of up to 20 S5 pixels. At nadir, a S5 pixel has a ground footprint of $500\times500$ square meters. Therefore, 20 S5 pixels span a distance of up to 10 km, which corresponds to, at most, about 4 grid cells. Distance between VNF flaring locations which are not adjacent and less than 4 grid cells away from the closest SLSTR-based location
may therefore arise from the clustering methodology. This is the case for 73 SLSTR-based locations and 1651 VNF locations.

The remaining 633 SLSTR-based locations and 4063 VNF locations were further apart. Of the distant locations in VNF but not considered as flaring by our methodology (4063 grid cells), slightly more than half (2222) had at least one SLSTR detection, of which almost all (1964) had at least one detection with the maximum retrieved temperature above 1500 K, see Figure 19. On the other hand, only 159 grid cells (out of the 2222 distant VNF locations that had at least one SLSTR-based
detection) had at least 5 high-accuracy SLSTR-based detections, see Figure 20.

We thus conclude that, at VNF distant locations, a lack of enough detections, and not the temperature, hinders our methodology to classify those grid cells as flaring locations. Since the overpasses are not that apart in time, cloudiness cannot account for this. The remaining factors affecting the detection opportunity are the overpass time and the swath. The difference in the

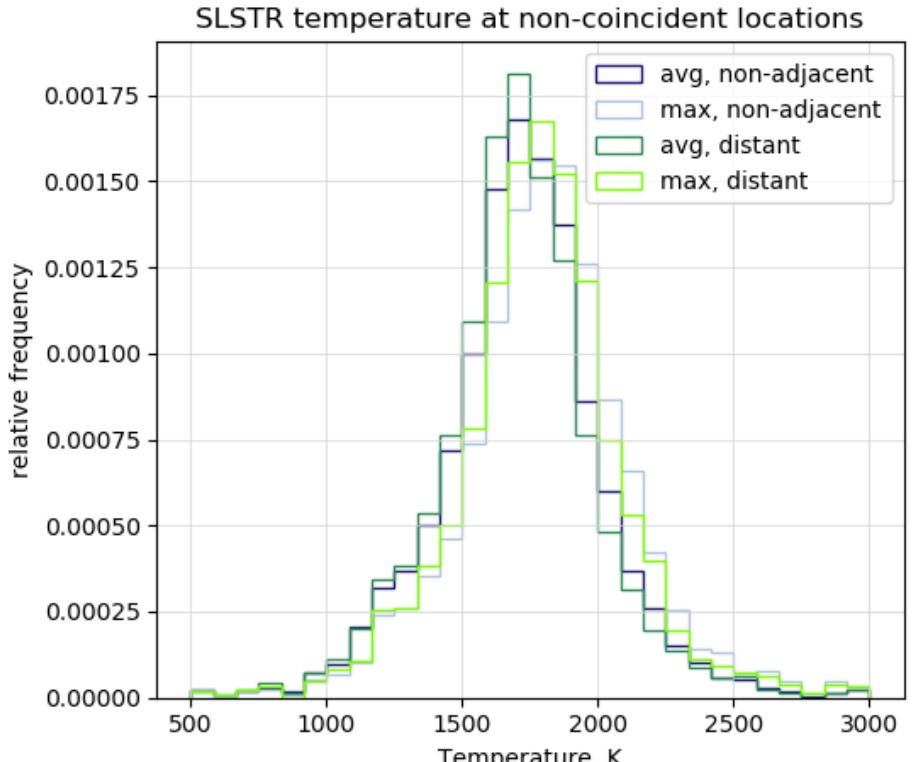

**Figure 19.** Average and maximum SLSTR-based temperature distributions at VNF VNF non-adjacent (the VNF flaring location is ≥ 2 grid cells away from the closest SLSTR-based flaring location) and distant (>4 grid cells away) locations. The similar temperature distribution between the different types of flaring sites supports the argument that the discrepancy in the number of flaring sites is not due to the temperature criterion, but to the observation criterion, which can be traced back to less observation opportunities.

overpass time between the sensors may play some role: the Suomi-NPP platform observes later during the night, thus possibly having more opportunities to detect gas flares at night. However, the effect of the different swaths of the instruments is likely more important. VIIRS has a much larger swath (3040 Km against 1420 Km for SLSTR) and has therefore more opportunities of detection. The detection opportunity is more important if the flare exhibits little activity and, indeed, the 4063 VNF distant

5   flaring locations account for just 8.7 BCM in the VNF inventory. The methodology presented here could be adjusted and the threshold in the number of high-accuracy observations lowered, but this would come at the cost of increasing false positives (see section 2).

The very large majority of the VNF-based activity was detected at locations coincident with SLSTR-based locations (87% at coincident locations, 92% at coincident or adjacent locations). 98% of the SLSTR-based activity was detected at coincident

10   or adjacent locations. This shows that both methodologies detect similar global levels of activity (151 BCM computed by VNF

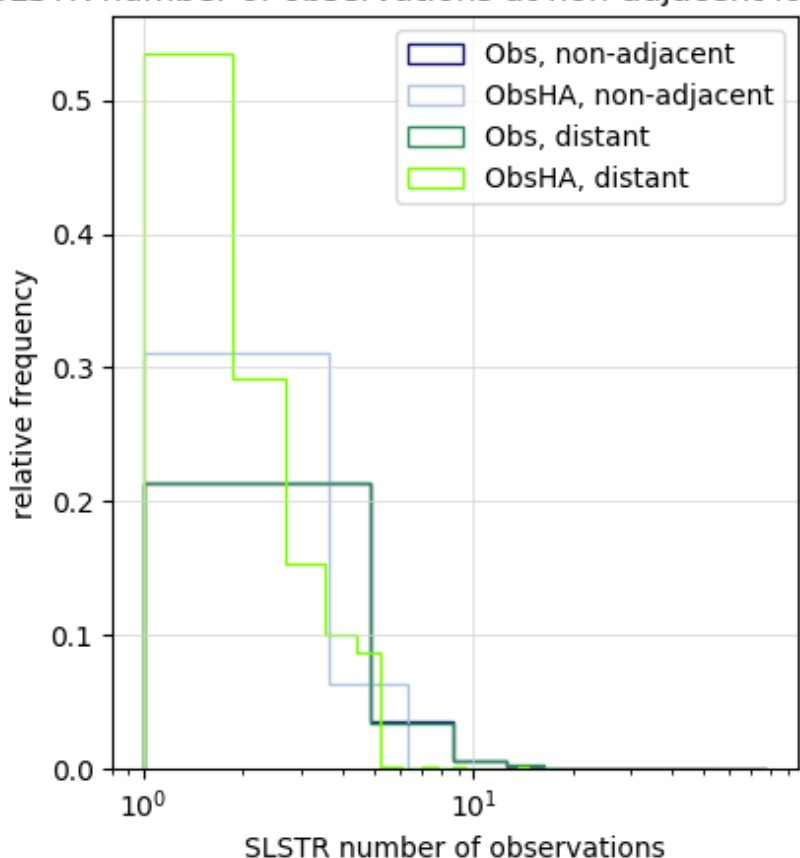

**Figure 20.** Number of SLSTR-based observations and high-accuracy observations distributions at VNF non-adjacent (the VNF flaring location is $\geq 2$ grid cells away from the closest SLSTR-based flaring location) and distant ($>4$ grid cells away) locations. The diminute number of observations at distant locations supports the argument that our methodology has a tendency to undersample low activity flares when compared to VNF. Our analysis shows that the undersampling is due to less observation opportunities by SLSTR, which can be traced back to a slighty earlier overpass time and a much smaller swath.

and 129 BCM for the present work) and this at the same locations, despite the large discrepancy in the number of flaring locations.

Table 2 shows that the difference in the computed global activity comes mostly from coincident and adjacent flaring locations, which indicates that the computation of the activity levels is more likely the source of the discrepancy (22 BCM globally) rather than the detection itself. One possibility is the scaling we use to obtain the annual activity from the individual detections (see Section 2, Equation 3 and Figure 2). Another source of discrepancy could be the lower temperature retrieved by our method, which considers the TIR for the Planck curve fitting.

Figure 21 shows the global distributions of the average retrieved flaring temperatures. The generally lower temperature in the present work can be traced back to the clustering methodology we take and was discussed in our previous paper (Caseiro et al., 2018). The use of the TIR channels in the dual Planck curve fitting may also lower the retrieved hot source temperature.

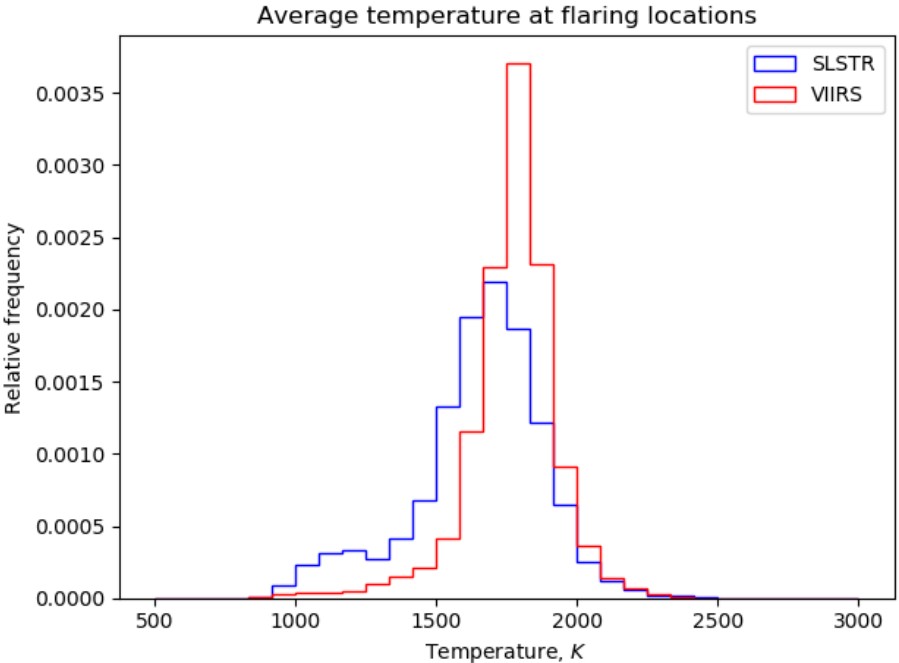

**Figure 21.** Comparison of the average retrieved temperature at the flaring locations for VIIRS Nightfire (locations shown in Figure 16 in red) and for SLSTR (locations shown in Figure 4 in blue). Relative frequency is used to ensure comparability independently of the total number of observations (6232 flaring locations for SLSTR, 10185 flaring locations for VIIRS Nightfire).

## 3.6 Comparison with the SWIR-radiance method

Fisher and Wooster (2018) developed a methodology to derive the radiative power from flares using a single SWIR channel, analogous to a previous methodology which used the MWIR radiance to derive the fire radiative power in the case of landscape fires (Wooster et al., 2005). The methodology is based on the near-constant ratio between the SWIR spectral radiance and the total emitted radiance at typical flaring temperatures (i.e. 1600–2200 K). The radiative power can be computed, using a sensor-specific optimized relationship and without the input of the emitter's temperature, with a theoretical precision of 13.6 %.

Fisher and Wooster (2019) used the SWIR-radiance methodology to derive a global time series of flaring activity using (A)ATSR and SLSTR data. For SLSTR, the data used spanned from May 2017 to May 2018. The identification of night-time thermal hotspots was conducted using a simple static threshold: pixels having a radiance above the instrumental noise level were considered as hot spots. The thermal hotspots are subsequently gridded onto a 1-arcminute global geographic grid (approximately 2 km). For SLSTR, a grid cell is considered as a flaring location if there were at least two different months with hotspots observations. The authors report 10428 locations, with a global distribution very similar to what we report here.

In order to compare the methodologies, we apply the Single-SWIR method to our detections and compare the output (RP) to our results. Since the BCM in both methodologies is derived from linear models, the differences seen in the RP comparison would be reproduced in a BCM comparison.

In order to compare the methodologies, we apply the SWIR-radiance approach developed for SLSTR's S5 SWIR channel by Fisher and Wooster (2019) to our results (197610 valid detections at flaring locations) and compare the output (RP). We limit the comparison to detections for which the retrieved temperature falls in the range 1600–2200  and which were limited to one S5 pixel in order to comply with the pre-requisites of the SWIR-radiance approach. The SWIR-radiance approach also allows to compute the activity. However, and since the BCM in both methodologies is derived from linear models, the differences seen in the RP comparison would be reproduced in a BCM comparison.

Figure 22 shows the comparison between the RP derived from the SWIR-radiance method and the Planck-curve fitting followed by the application of the Stefan-Boltzmann equation. There is a good correlation, although the SWIR-radiance methodology outputs radiative power values about 25 % lower than those from this work. The trend is more recognizable at lower temperatures, which might indicate that the relationship between the SWIR spectral radiance and the total emitted radiance deviates more from linearity at lower temperatures.

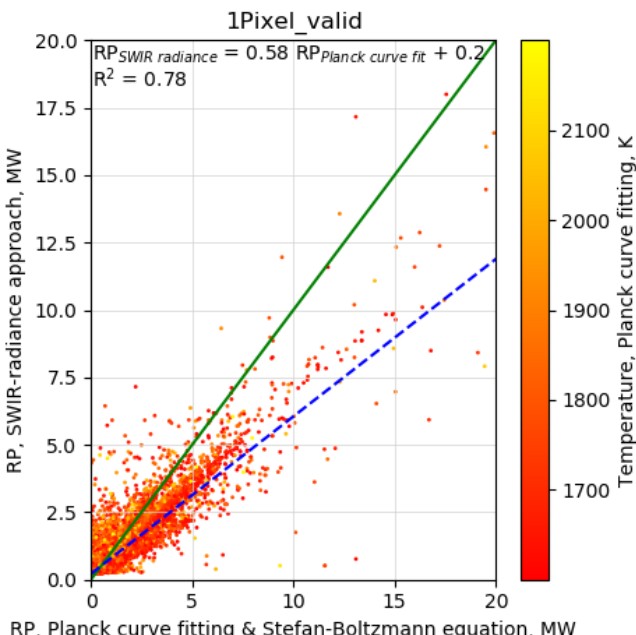

**Figure 22.** Comparison between the SWIR-radiance method (Fisher and Wooster, 2018) and the SLSTR-based Planck curve fitting. The method derived by Fisher and Wooster (2018) and applied to SLSTR by Fisher and Wooster (2019) was applied to the hotspots detected by our method at flaring locations. Only detections with one pixel and whose retrieved temperature was in the range considered valid for the SWIR-radiance method (between 1600 and 2200 K) were considered. The continuous line represents the 1:1 relationship. The dashed line is the regression line between the methods.

# 4 Conclusions

In this work we apply a previously described hot spot detection and characterisation procedure (Caseiro et al., 2018) to the 2017 observations of Copernicus Sentinel-3A's Sea and Land Surface Temperature Radiometer (SLSTR) instrument. In our previous paper, only persistency was used to discriminate gas flares from other sources of heat emissions. Here, we refine the discrimination by updating the persistency criterion and introducing a temperature criterion, two characteristics of gas flares. The development of the refined discrimination strategy is reported. Validation through referencing with high-resolution images yields a detection accuracy of 85±11%, with a commission error of 3±1%.

Our methodology detects 6232 flaring sites worldwide in 2017. Over half of these are located in five countries only: Russia, the United States, Iran, China and Algeria.

Additionally, we calculate the volume of flared gas based on a previously observed and published linear relationship between the flared volume and observed flare radiative energy. Subsequently, estimates of the associated black carbon (BC) emissions are calculated with a new dynamic emission factor parameterization.

The best estimate of flared gas at the detected flaring sites is calculated by applying a latitude-depending scaling factor that roughly describes for the variation in observation opportunities due to cloud cover and overpass frequency. The confidence interval is specified with the following bounds: The upper bound is computed as if the flare would constantly burn for the whole year. The lower bound is computed as if the flare would burn on days with hot spot detections only. The spread of the confidence intervals and the accuracy of the best estimates could certainly be reduced in the future with a more sophisticated correction for different observations opportunities that accounts for the actual number of cloud-free and solar uncontaminated overpasses of each individual flaring site.

Our analysis yields a result of 129 billion cubic metres (BCM) of gas flared in 2017 (best estimate) with a confidence interval of [35,419] BCM. Only four countries are responsible for half of the global gas flaring activity: Iraq, Iran, Russia and Algeria. The most active flaring location is located in Venezuela and burned 0.62 BCM in 2017.

A comparison with the VIIRS Nightfire dataset shows that the number of flaring sites assessed here is significantly lower (6232 vs. 10185). Three main reasons can lead to the discrepancy: (1) Small geolocation errors may result flares being assigned to adjacent grid cells. (2) The clustering methodology employed for SLSTR may result in an assignment of gas flare arrays spanning several grid cells to only one central grid cell. (3) The relatively stringent persistency criterion of the SLSTR method certainly excludes some infrequently operation gas flares. The former two reasons result in a mismatch between the SLSTR-based and VNF locations of gas flares but the aggregated budgets by country are not affected. The stringency of the persistency criterion might affect 4063 of the 10185 VNF locations and these have a relatively little contribution of 8.7 BCM to the annual total of 151 BCM in VNF. The incorporation of Sentinel 3B in our system would lead to more observation opportunities. The inclusion of this feature and an analysis for the following years is pending and we expect it to provide substantially more coincidences between the two datasets. The complementarity and consistency of both products indicate that a merged VIIRS/SLSTR product would provide substantial improvement of the global flaring activity and emissions.

We also compare our activity estimates to the SWIR-radiance method. Both methodologies show a good agreement for high-temperature detections, but a constant under-estimation by the SWIR-radiance method for lower temperature detections.

We further quantify the black carbon (BC) emissions due to gas flaring. We assume that temperature is an indication of the completeness of combustion and use the retrieved flame temperature to determine the emission factor for each single detection, bounded by the range of previously published emission factors. We assume that a temperature close to the adiabatic flame temperature corresponds to the lower bound of the emission factor range considered, while the lowest temperature corresponds to the upper bound. The minimum, maximum and best estimates for the BC emissions are computed in the same way as the flared volume. Our resulting best estimate for the emission of BC to the atmosphere by gas flaring in 2017 is 73 Gg with a confidence interval of [20,239] Gg, with Iraq, Iran, Russia and Algeria being responsible for roughly one half of those emissions. The most active flaring location, which is also the one with the highest emissions, is estimated to have yearly emission of 0.35 Gg of BC in 2017.

This study shows that the SLSTR instruments on-board the Copernicus Sentinel-3 satellites are well suited for the quantitative characterization of the gas flares in terms of flame temperature, size and radiative power, as well as BC emissions. This will allow to increase the detection opportunity for gas flares in an observation system that comprises SLSTR and VIIRS instruments.

## 5 Data availability

The flaring activity and the related black carbon emissions product are available with $0.025° \times 0.025°$ resolution on the Emissions of atmospheric Compounds and Compilation of Ancillary Data (ECCAD, DOI 10.25326/19 (Caseiro and Kaiser, 2019)) web site (https://eccad3.sedoo.fr) for use in, e.g., atmospheric composition modelling studies.

*Author contributions.* Conceptualization, AC and JWK; Methodology, AC, JWK and GR; Software, AC and DL; Validation, AC, JWK and BG; Formal Analysis, Investigation & Resources, AC and BG; Data Curation, AC, BG, DL, GR and JWK; Writing–Original Draft Preparation, AC; Writing—Review & Editing, BG, JWK & GR; Visualization, AC; Supervision, Project Administration & Funding Acquisition, JWK and GR.

5    *Competing interests.* The authors declare that there are no competing interests.

*Acknowledgements.* The data was produced in the project "GFS3 - Identification of gas flares and quantification of their emissions using Sentinel-3 SLSTR" funded by the German Federal Ministry for Economic Affairs and Energy (BMWi, Bundesministerium für Wirtschaft und Energie) under contract number FKZ 50EE1339. DL and GR also acknowledge support by BMWi through the ZIM project FireSense (FKZ 16KN052420). We acknowledge Daniel Fisher and Chris Elvidge for their comments. We acknowledge Mariapia Faruolo and the
10    remaining two anonymous referees for their reviews.

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
