# Peer review of "Gas flaring activity and black carbon emissions in 2017 derived from Sentinel-3A SLSTR"

_Earth System Science Data, 2019_

## Referee Comment (RC1) · Anonymous Referee #1 · 28 Aug 2019

Dear authors,

I reviewed the manuscript entitled "Gas flaring activity and black carbon emissions in 2017 derived from Sentinel-3A SLSTR". I found the basic idea interesting, considering the great potential the Sentinel missions will provide in the future, but in my opinion a hard work has to be done for make the analysis more consistent. The main doubt is you do not well know (and so you incorrectly use) the VNF dataset. The part inherent to the flaring sites detection is indeed confused. The topic of your paper is the characterization of flaring sites, in terms of gas flared volumes and black carbon emissions estimates. First, you improve the performances of your pervious work (Caseiro et al 2018) in detecting flaring sites, adding a temperature filtering. When you compare your results with VNF, you first use the 2012 VNF outputs (why not the 2017?) and then you take into account the combustion sources (https://ngdc.noaa.gov/eog/viirs/download_viirs_fire.html) identified by VNF instead of the flaring sites available at https://www.ngdc.noaa.gov/eog/viirs/download_global_flare.html for 2017 (the year of your analysis). I think it is a forcing applying the criteria developed in this work for SLSTR to select among the VNF combustion sources the flaring sites. The latter are directly provided by NOAA at https://www.ngdc.noaa.gov/eog/viirs/download_global_flare.html.

Below my suggestions/corrections.

**Abstract**
- We calculate the global flared gas volumes and black carbon emissions in 2017 by (1) applying (1) a previously developed hot spot detection and characterisation algorithm to all observations of the SLSTR instrument on-board the Copernicus 5 satellite Sentinel-3A in 2017 and (2) applying newly developed filters for identifying gas flares and corrections for calculating both flared gas volumes (BCM) and black carbon emission (g/m$^3$) estimates.
- The comparison of our results with those of the VIIRS Nnightfire data set indicates a good fit between the two methods.
- Please, remove the space at the beginning of the bracket ( https://eccad3.sedoo.fr/#GFlaringS3, DOI 10.25326/19 (Caseiro and Kaiser, 2019))

**Introduction**
- Please, put the dot after the references: or convert the gas. (Rahimpour and Jokar, 2012; Emeka Ojijiagwo et al., 2016). This is the first case, I found many others in the paper.
- Improvements of flare gas recovery systems haves been recommended …
- GF also impacts the environment on a wider scale through the emission of pollutants and greenhouse gases like carbon dioxide (CO$_2$), carbon monoxide, black carbon (BC)…
- Of particular importance is also the black carbon (BC) emission emitted by GF. BC is a known carcinogen (Heinrich et al., 1994) as well as a short-lived climate forcer (IPCC, 2013). BC strongly aeffects environments such …
- Satellite remote sensing has been utilized for regional and global identification and characterization of GF. (Casadio et al., 2012b, a; Anejionu et al., 2014; Faruolo et al., 2014; Chowdhury et al., 2014; Anejionu et al., 2015; Faruolo et al., 2018). The most prominent system is NOAA's VIIRS (here add NOAA acronym Visible Infrared Imaging Radiometer Suite) Nightfire (VNF) dataset (see https://ngdc. noaa.gov/eog/viirs/download_viirs_fire.html), developed by Elvidge et al. (2013, 2016) for the detection and characterization of combustion sources based on previous work (Elvidge et al., 2001, 2007, 2009, 2013) and leading to a globally consistent survey of gas flaring volumes available extending back to 2012 (https://www.ngdc.noaa.gov/eog/viirs/download_global_flare.html).
- We recently published an adaptation and extension of the VNFVIIRS Nightfire algorithm with which observations of the SLSTR instrument (Sea and Land Surface Temperature Radiometer) instrument on-board the Sentinel-3A satellites havecan been analysed, too (Caseiro 30 et al., 2018).
- The main advantages of using our hot spot detection and characterisation algorithm lie in the ability to detect and quantify smaller flares and the foreseen long term data availability from the series of Sentinel-3 satellites in the Copernicus program. Additionally, SLSTR observations (night-time overpasses at 10:00 PM) complement those of VIIRS (1:30 AM) by filling observation gaps in the time

series. I think the unique advantage your algorithm seems to offer, when compared to VNF, is its capability to identify smaller flares. Regarding the data continuity, also VIIRS is actually onboard two satellites (Suomi NPP and JPSS-1) and will also be flown on the JPSS-2 (launch in 2021), -3 (2026) and -4 (2031) satellite missions. You can rephrase this sentence, pointing out the potential of these algorithms, the possibility of integrating them as well as of continuously monitoring the phenomenon thanks to the long design life of satellite missions.

- Here, we describe a new dataset of global gas faring volume (BCM) and BC emissions (g/m3), which we have derived from all Sentinel-3A SLSTR observations in 2017. In detail, chapter 2 describes newly developed methods for identifying gas flares among the observed hot sources, correcting for intermittent observations opportunities, and dynamically determining appropriate BC emission factors from the observations. The results  are presented in Chapter 3, the conclusions are summarised in Chapter 4.
- While in principle the methodology used is based on the VNF
- We already tested the method using oil and/or gas producing regions within a limited timespan and compared the results to the VNF

**2.1 Hot spot detection and characterization**
Figure 1 should be improved, explaining the GF filter.

**2.2 Hot spot classification**
**2.2.1 Volcano filter**
- The data were filtered
- Many volcanoes do not consist of a single edifice,  many individual eruptive fissures through which lava erupts may be present in a volcanic field (Siebert et al., 2010).

**2.2.2 Discrimination of gas flares from other industrial hot sources**
This paragraph is not completely clear. You are searching for a criterion to use for accurately detecting flaring sites. The starting point is your algorithm (Caseiro et al., 2018), to which you add a temperature filtering. I do not understand how you use the works of Elvidge et al. (2016) and Liu et al. (2018) in the definition of the temperature criterion. To this aim, you test several subsets. Can you explain what are these subsets? They are 8? They correspond the 8 columns in Table 1? Besides, I expected $n_{Obs}$ was greater than $n_{ObsHA}$. Probably, it is more correct to use ≥ than >.

**2.3 Determination of flared volumes and black carbon emissions**
- Please, explain the terms BCMmin, BCMmax, BCMbest in this order, to facilitate the comprehension.
- The emissions of black carbon (BC) from gas flares are estimated using reported emissions factors (EF). It could be useful to specify the formulation applied for their computation.
- GAINS: please, extend the acronym.
- You define flaring site a site with a temperature above 1500K. Why do you compute the EFs for lower temperatures?
- With this methodology we estimate a wide range of possible activity (BCM) and BC emissions ($g/m^3$)
- Can you better explain this sentence, please? I do not understand it: "We conservatively assume that this range of possibilities represents 6 × σ, and report the uncertainty of the best estimates as 1 × σ".

**3. Results**
**3.1 Hot spots and flaring sites**
I have concerns about this section. Your paper focuses on gas flaring, the previous one (Caseiro et al., 2018) on hotspots. For this reason, you can join Figures 4, 5, 6 using three colors for discriminating hotspots, high confidence hotspots and flaring sites. Besides, I do not understand why you compare the SLSTR global detections for 2017 with the VNF in 2012. The VNF data for 2017 are available; you indeed use them in section 3.3.

- Russia (985) and the United States (917) are the countries with the highest number of flaring locations (Figure 7).
- The time series of the cumulative number of the high accuracy observations for the most active flaring location (in Venezuela, see Section 3.4) is shown in Figure 8. It shows flaring activity throughout the year. In my opinion, it is not useful and interesting. Remove Figure 8.

I think 3.2 and 3.3 are subsections of 3.1: they become 3.1.1 and 3.1.2.
- Figures 10, 11 and 12 are not useful, in my opinion they could be removed. You can indeed add before Figure 9 and Figure 13, respectively, a global map (in color scale) showing the temperatures and RP values for the 6232 sites.
- Figure 9. Distribution of the average retrieved  temperature (K) for the flaring locations
- The average temperature at the flaring locations approximately ranges from 950 K to 2250 K. This is slightly lower than the range reported by Liu et al. (2018) (please, can you specify the values) who used VIIRS Nightfire data, as expected from our previous study (Caseiro et al., 2018). It confirms the bi-modal distribution with modes around 1750 K and 1200 K that is has also been observed by VIIRS.
- The section "Comparison with VIIRS Nightfire" should be modified. As before explained, being the focus of your work the gas flaring, you should compare your results with the VNF flaring sites (available at https://www.ngdc.noaa.gov/eog/viirs/download_global_flare.html), avoiding to select these sites among the VNF combustion sources applying the criteria used for SLSTR.
- You never cite Figure 14 in the paper. The figure is not useful, as figures 10-12.

**3.4 Flared volumes (new 3.2)**
As before, you should use BCM data available at https://www.ngdc.noaa.gov/eog/viirs/download_global_flare.html for the comparison with your estimates in 2017. It would be interesting the map of the global distribution of BCMbest. In Figure 20 you could add the distribution derived by the VNF data elaboration.

**3.5 BC emissions (new 3.3)**
As for BCM, you can add a global map of BC emissions.

**Conclusions**
To reorganize based on new suggested analyses. In any case:
- The sentence "We present a new gas flaring discrimination procedure, based on two characteristics of gas flares: persistence and temperature" is not correct. This procedure is not new, being the one most used to identify gas flares. Respect to your methodology, you simply add a temperature filtering to improve the detection of flaring sites.
- "Additionally to the detection we present a way to assess the volume of flared gas": is not true. You apply a widely declared model developed by Elvidge et al (2016) to compute monthly flared volumes, adding a scaling factor, which takes into account the operation time of the sites.

---

## Short Comment (SC1) · 11 Sep 2019

Review of "Gas flaring activity and black carbon emissions in 2017 derived from Sentinel-3A SLSTR" The paper attempts to locate all the active gas flares of 2017 and estimate their flare gas volumes using nighttime data collected by the Sea and Land Surface Temperature Radiometer (SLSTR) instrument flown on-board the Copernicus satellite Sentinel-3A. The basic detection algorithm for the individual nights of data follows the VIIRS nightfire (VNF) method and appears to be solid. But the steps used to go from the individual nights of data to the annual summary are questionable and should be revisited: 1. The SLSTR results found 6232 flaring sites in 2017. This compares to over 10,000 flares reported by the VIIRS nightfire team for 2017 (https://eogdata.mines.edu/download_global_flare.html). 2. Many flares are intermit-

tent. The nightly flare detection data does not contain sufficient information to account cloud and solar contamination effects that could effect the annual flared gas volume calculation. Hence the annual characterization of flared gas volume should calculate the "duty cycle" or "percent frequency of detection" for each flaring site. The VNF team makes the calculation based on flare detection numbers in the set of nighttime cloud-free observations made of the site during the year. Because the nightly VNF product only contains the detections – the annual analysis includes an inventory of the cloud state (cloudy or clear) for the nights lacking VNF collection that are free of solar contamination. The VNF method excludes both sunlit and cloudy observations in the calculation of flaring site duty cycle. The method reported in this paper (section 2.3 and Figure 2) is woefully inadequate and appears to have resulted in a drastic under-estimation of annual flared gas volume in Russia. I suspect that the method in Figure 2 does not account for solar contamination outages during summer months – as shown below with VNF for a flare in northern Siberia.

3. The paper lacks detail on the method used to discriminate clear versus cloudy observations. In addition, the paper makes several assertions that should be rechecked: 4. The paper makes several claims that the "SLSTR-based methodology is able to detect smaller gas flares". No evidence is presented to back up this claim. 5. The paper state that the VNF product only uses a single shortwave infrared channel. This was the case for early VNF data. However, from January 2018 forward VNF from two satellites has included two SWIR bands. My recommendation is that the paper undergo major revision and a second round of peer review prior to publication. The authors should make a specific comparison against the VNF product from 2017 to better understand difference between the SLSTR and VIIRS flaring sites and flared gas volumes. Are there specific geographic regions where one system detects more flares or more flared gas volumes? Since the instruments and detection algorithms are so similar, the authors should figure out the reason behind the discrepancies. To make a direct comparison of the combustion source detection limits with VNF, the authors can follow the methods outlined in https://www.mdpi.com/2072-4292/11/4/395. This

crosschecking with VNF could lead to major improvements in the gas flaring results from SLSTR and a far better paper.

Please also note the supplement to this comment:
https://www.earth-syst-sci-data-discuss.net/essd-2019-99/essd-2019-99-SC1-supplement.zip
* * *
Interactive
comment

[Figure]

**Fig. 1.** Temporal profile VNF SWIR radiance for a flare in Siberia. Each year there is an outage period due to solar contamination.

---

## Short Comment (SC2) · 13 Sep 2019

Gas flaring activity and black carbon emissions in 2017 derived from Sentinel-3A SLSTR.

In this paper the thermal anomaly detection and characterisation algorithm developed in Casiero et al. (2018), based on NightFire, is applied to Sentinel-3 SLSTR data with the objective of evaluating global gas flaring radiant heat output and the associated estimates of black carbon emissions. The resulting thermal anomaly outputs, generated for 2017, are subjected to various filters to partition flaring and non-flaring anomalies, with the most crucial step being the application of a persistency test of more than five observations in the 12 month evaluation period in addition to a minimum temperature

limit of 1500 K, which must be exceeded. A normalisation process is then applied to the detected gas flaring sites to account for differences in sampling opportunities to generate an adjusted measure of radiant heat output for each flaring site. Black carbon emissions estimates are then generated from the adjusted radiant heat estimates using appropriate emission factors. Finally comparison against various other datasets are made to assess the validity of the SLSTR generated datasets.

Whilst in general the paper is reasonably well written, there is a potentially significant flaw in the gas flare characterisation approach which must be investigated and addressed before publication can be recommended. In Elvidge et al. (2016) and Fisher et al. (2019) it is demonstrated that a substantial proportion of global gas flaring radiant heat output arises from a small subset of flaring sites global ($\sim$50% of output comes from 5-10% of all flares). Some of these flares are extremely radiant such as the Punta de Mata site in Venezuela identified by the authors (and elsewhere), and they attribute 0.623 BCM of flared gas to this site in 2017. In comparison, Elvidge et al (2016) identified that 1.13 BCM of gas was flared at this site in 2012. The discrepancy between these two values is concerning, particularly as Venezuela has been shown to have had a very large increase in gas flare radiant heat output since 2012 (Fisher et al., 2019), and given the characteristics of gas flaring (e.g. most radiant heat being produced by a small subset of sites) one would expect that the most active flaring site in Venezuela to have show at least some increase, and not the reported decrease.

I think that this result may be arising from a potential issue with the channels used by the algorithm. In Elvidge et al. (2019) the issue of saturation in the (M11) 2.2 $\mu$m and (M12) 3.7 $\mu$m VIIRS channels is identified and I would expect that a similar issue is occurring with SLSTR, particularly given the enhancement in pixel resolution from 750 m2 to 500 m2. The channels employed in the Casiero et al. (2018) for the estimation of gas flare radiant heat output are S5 (1.6 $\mu$m), S6 (2.2 $\mu$m) and S7/F1 (3.7 $\mu$m). Given its specifications, the S7 channel likely saturates on a regular basis over gas flares, and this is identified, as is the potential for using the F1 channel when this occurs.

[Figure]

However, no assessment of the saturation characteristics of the SWIR channels is made, and whilst the S5 channel likely does not saturate, the same cannot be said for the S6 channel. Furthermore, being a relatively new instrument the performance of the F1 channel has not been evaluated, and it needs to be demonstrated that it reaches the specified dynamic range performance levels (the same can be said for the SWIR channels). If any of the channels are saturating then the effects on the retrieved radiant heat can be significant as shown in Elvidge et al. (2019), and in the current configuration of the algorithm used in this paper these saturation events may well be being missed. I expect, given that a significant proportion of radiant heat output arises from a subset of highly radiant flaring sites, the impact of saturation on the reported global total of radiant heat output and in turn BCM could well be significant and must be explored.

To do so I would recommend (1) evaluating the Planck curves of the most radiant flare sites identified globally and check the deviations of the various spectral observations from the curves that might indicate saturation of specific channels, giving at least an indication of saturation effects on SLSTR. (2) I would recommend applying the single channel radiant heat estimation approach of Fisher and Wooster (2018) developed in part for application to the S5 channel of the SLSTR sensor and see if any significant differences are observed, this would be a very straightforward comparison. (3) I would then suggest that if a discrepancy for these larger flaring sites is found that the S3 channel is included as an additional constraint for these very large flares to try and improve the radiant heat estimates. (4) It would be likely be useful to compare also against flare counts from Fig. 7 in Fisher and Wooster (2019) in addition to the NightFire comparison.

Lastly, some key references are missing from the paper and should be included:

Anejionu, O.C., 2019. Rationale, historical developments and advances in remote sensing of gas flares. International Journal of Remote Sensing, 40(17), pp.6700-6719.

Elvidge, C.D., Zhizhin, M., Baugh, K., Hsu, F.C. and Ghosh, T., 2019. Extending night-time combustion source detection limits with short wavelength VIIRS data. Remote Sensing, 11(4), p.395.

Fisher, D. and Wooster, M., 2018. Shortwave IR Adaption of the Mid-Infrared Radiance Method of Fire Radiative Power (FRP) Retrieval for Assessing Industrial Gas Flaring Output. Remote Sensing, 10(2), p.305.

Fisher, D. and Wooster, M.J., 2019. Multi-decade global gas flaring change inventoried using the ATSR-1, ATSR-2, AATSR and SLSTR data records. Remote Sensing of Environment, 232, p.111298.

Other references referred to here:

Caseiro, A., Rücker, G., Tiemann, J., Leimbach, D., Lorenz, E., Frauenberger, O. and Kaiser, J., 2018. Persistent Hot Spot Detection and Characterisation Using SLSTR. Remote Sensing, 10(7), p.1118.

Elvidge, C. D., Zhizhin, M., Baugh, K., Hsu, F.-C., and Ghosh, T.: Methods for Global Survey of Natural Gas Flaring from Visible Infrared Imaging Radiometer Suite Data, Energies, 9, 14,

---

## Referee Comment (RC2) · Anonymous Referee #2 · 2 Jan 2020

The paper on gas flaring activity and black carbon emissions in 2017 based on Sentinel by Caserio et al presents an important dataset that is useful and relevant for the scientific community. However, at this stage, the paper has several shortcomings in the methodology and description and needs further revision. In addition to comments from reviewer 1, I point out a few instances where the paper needs to be further improved:

1. The methodology section is too brief and the reader needs to go through many other publications to gain even an overview of the current one. This is inconvenient for the general readers. E.g. in line 12 of page 3, instead of directly pointing to Caserio et al. (2018), just very briefly give an overview of the salient features of the algorithm. Similarly, for line 19 of page 3, explain why analyzing he cluster of contiguous hot pixels advantageous than the spatial maxima method. Additionally, what could be the

disadvantages of this method and how are they taken care of...e.g. What are the possibilities that pixels representing different intensities taken as an average may lead to an overall over or underestimation for a grid or lead to a mixing of two very different signals.

2. Section 2.2.2 : state the background, advantages and disadvantages for using he Equations 1 and 2.

3. Section 2.3: The BC estimation formula seems too oversimplified. E..g. equation 4 does not take into account the different flares leading to different amounts of BC emissions over the year. This may be a reason leading to underestimation w.r.t. other inventories...I believe authors should instead try something like a weighted average or a more representative method to arrive at a better estimate.

4. The diagrams need to be more complete by themselves e.g. y label missing in fig 11, 12 (only writing it within caption is not sufficient), cbar label missing in fig 10

Minor comments:

Page 2 Line 11: Contributions to what e.g. CO2 equivalent?

Page 2 Line 14: Replace 'emitted' by 'from'

Page 3 Line 4: Replace 'chapter' by 'section'

Page 21 line 9: what is Upstram?

Page 24 Line 4: Replace 'allow' by 'allows'

---

## Author Comment (AC1) · 29 Apr 2020

Dear reviewer,

Thank you very much for your valuable remarks, comments and suggestions. We find that by answering your questions and comments, and by following your suggestions, we have improved the readability of our work.

We start by answering the points raised in your general comment and then proceed with the section-specific suggestions/corrections.

In this document, your original comments are framed by a box and our answer follows.

**Answer to the general comment:**

> First, you improve the performances of your pervious work (Caseiro et al 2018) in detecting flaring sites, adding a temperature filtering.

We indeed complete our previous work with a filtering procedure which is based on the analysis of the temperature time series retrieved at the location of a given detection (the maximum temperature must be larger than 1500K) and on the persistence of the signal at that location (more than 5 quality detections per year).

> When you compare your results with VNF, you first use the 2012 VNF outputs (why not the 2017?) and then you take into account the combustion sources (https://ngdc.noaa.gov/eog/viirs/download_viirs_fire.html) identified by VNF instead of the flaring sites available at https://www.ngdc.noaa.gov/eog/viirs/download_global_flare.html for 2017 (the year of your analysis). I think it is a forcing applying the criteria developed in this work for SLSTR to select among the VNF combustion sources the flaring sites. The latter are directly provided by NOAA at https://www.ngdc.noaa.gov/eog/viirs/download_global_flare.html.

Our first decision was to use only published data for the activity and emissions comparisons and the most up-to-date VNF data for the characterisation (in our case, temperature), after applying a similar procedure as the one used in our work (i.e. gridding). We accept the suggestion and include the 2017 VNF temperature and activity data in our analysis. See section 3.5 for the comparison with the 2017 VNF dataset.

**We now answer the section-specific suggestions/corrections:**

> **Abstract**
> - We calculate the global flared gas volume and black carbon emissions in 2017 by  applying (1) a previously developed hot spot detection and characterisation algorithm to all observations of the SLSTR instrument on-board the Copernicus 5 satellite Sentinel-3A  and (2)  newly developed filters for identifying gas flares and corrections for calculating both flared gas volumes (BCM) and black carbon emission (g/m³) .
> - The comparison of our results with those of the VIIRS Nightfire data set indicates a good fit between the two methods.
> - Please, remove the space at the beginning of the bracket ( https://eccad3.sedoo.fr/#GFlaringS3, DOI 10.25326/19 (Caseiro and Kaiser, 2019))

All your recommendations were followed in the updated manuscript, except the units for the BC emissions, which we kept as mass (g).
* * *
**Introduction**
- Please, put the dot after the references: or convert the gas. (Rahimpour and Jokar, 2012; Emeka Ojijiagwo et al., 2016). This is the first case, I found many others in the paper.
- Improvements of flare gas recovery systems haves been recommended ...
- GF also impacts the environment on a wider scale through the emission of pollutants and greenhouse gases like carbon dioxide $(CO_2)$, carbon monoxide, black carbon (BC)...
- Of particular importance is also the black carbon (BC) emission emitted by GF. BC is a known carcinogen (Heinrich et al., 1994) as well as a short-lived climate forcer (IPCC, 2013). BC strongly aeffects environments such ...
- Satellite remote sensing has been utilized for regional and global identification and characterization of GF. (Casadio et al., 2012b, a; Anejionu et al., 2014; Faruolo et al., 2014; Chowdhury et al., 2014; Anejionu et al., 2015; Faruolo et al., 2018). The most prominent system is NOAA's VIIRS (here add NOAA acronym Visible Infrared Imaging Radiometer Suite) Nightfire (VNF) dataset (see https://ngdc. noaa.gov/eog/viirs/download_viirs_fire.html), developed by Elvidge et al. (2013, 2016) for the detection and characterization of combustion sources based on previous work (Elvidge et al., 2001, 2007, 2009, 2013) and leading to a globally consistent survey of gas flaring volumes available extending back to 2012 (https://www.ngdc.noaa.gov/eog/viirs/download_global_flare.html).
- We recently published an adaptation and extension of the VNFVIIRS Nightfire algorithm with which observations of the SLSTR instrument (Sea and Land Surface Temperature Radiometer) instrument on-board the Sentinel-3A satellites havecan been analysed, too (Caseiro 30 et al., 2018).
* * *
- [last comment in the above box] Since the methodology is applicable to all (to date 2) the SLSTR instruments, we prefer to keep the reference to the Sentinel-3 satellites in the plural.
- All the other recommendations were followed in the updated manuscript.
* * *
- The main advantages of using our hot spot detection and characterisation algorithm lie in the ability to detect and quantify smaller flares and the foreseen long term data availability from the series of Sentinel-3 satellites in the Copernicus program. Additionally, SLSTR observations (night-time overpasses at 10:00 PM) complement those of VIIRS (1:30 AM) by filling observation gaps in the time series. I think the unique advantage your algorithm seems to offer, when compared to VNF, is its capability to identify smaller flares. Regarding the data continuity, also VIIRS is actually onboard two satellites (Suomi NPP and JPSS-1) and will also be flown on the JPSS-2 (launch in 2021), -3 (2026) and -4 (2031) satellite missions. You can rephrase this sentence, pointing out the potential of these algorithms, the possibility of integrating them as well as of continuously monitoring the phenomenon thanks to the long design life of satellite missions.
- Here, we describe a new dataset of global gas faring volumes (BCDM) and BC emissions (g/m3), which we have derived from all Sentinel-3A SLSTR observations in 2017. In detail, cChapter 2 describes newly developed methods for identifying gas flares among the observed hot sources, correcting for intermittent observations opportunities, and dynamically determining appropriate BC emission factors from the observations. The results of applying the hot source detection and characteristion algorithm plus the newly developed methods to all SLSTR observations of 2017 are presented in Chapter 3, the. Finally, our conclusions are summarised in Chapter 4.
- While in principle the methodology used is based on the Nightfire algorithm developed for VIIRSVNF
- We already tested the method using oil and/or gas producing regions within a limited timespan and compared the results to the VNFVIIRS Nightfire
* * *
All your recommendations were followed in the updated manuscript, except the units for the BC emissions, which we kept as mass (g). Regarding the first comment of the box above, we have rephrased the idea focusing the complementarity of the instruments and the methods.

**2.1 Hot spot detection and characterization**
Figure 1 should be improved, explaining the GF filter.

**2.2 Hot spot classification**
**2.2.1 Volcano filter**
- The data were filtered
- Many volcanoes do not consist of a single edifice,  many individual eruptive fissures through which lava erupts _may be present in a volcanic field._ (Siebert et al., 2010).

We have updated the manuscript following all these recommendations.

**2.2.2 Discrimination of gas flares from other industrial hot sources**
This paragraph is not completely clear. You are searching for a criterion to use for accurately detecting flaring sites. The starting point is your algorithm (Caseiro et al., 2018), to which you add a temperature filtering. I do not understand how you use the works of Elvidge et al. (2016) and Liu et al. (2018) in the definition of the temperature criterion. To this aim, you test several subsets. Can you explain what are these subsets? They are 8? They correspond the 8 columns in Table 1? Besides, I expected $n_{Obs}$ was greater than $n_{ObsHA}$. Probably, it is more correct to use $\geq$ than $>$.

We have updated the caption of Table 1 with more detail:

**Table 1.** User's accuracy (UA, %) and commission error (C, %) of the hot spot discrimination strategies considered. $n_{Obs}$ is the number of hot spot detections within a grid cell, $n_{ObsHA}$ is the number of high-accuracy hot spot detections within a grid cell, $T_{min}$ is the minimum temperature retrieved among all the hot spots detected within a grid cell, $T_{max}$ is the maximum temperature retrieved among all the hot spots detected within a grid cell, $n_{cells}$ is the number of grid cells that comply to the thresholds. In order to discriminate gas flares from other hot spots we discriminate hot spots based on their persistency ($n_{Obs}$ and $n_{ObsHA}$) and on their temperature time series ($T_{min}$ and $T_{max}$). We have tried 8 combinations (discrimination strategies) of thresholds on those variables. Each column represent a tested discrimination strategy. For each of the 8 combinations, we examine high-resolution imagery for 100 random onshore locations (800 in total) in order to verify the presence of a gas flare. The goal is to maximize user's accuracy (UA) and minimize commission error (C) while minimizing the omission error (here, the variation in $n_{cells}$ is used as a proxy). The discrimination strategy #5 was selected as the most suitable.

| combination | #1 | #2 | #3 | #4 | #5 | #6 | #7 | #8 |
|---|---|---|---|---|---|---|---|---|
| $n_{Obs} >$ | – | 3 | 4 | – | – | – | – | – |
| $n_{ObsHA} >$ | 2 | 2 | 2 | 5 | 5 | 5 | 7 | 7 |
| $T_{min}\ (K) >$ | 1000 | 1000 | – | – | – | – | – | – |
| $T_{max}\ (K) >$ | 1400 | 1400 | 800 | 1200 | 1500 | 1800 | 1200 | 1500 |
| $n_{cells}$ | 6733 | 5872 | 9469 | 6817 | 6232 | 5485 | 5527 | 5129 |
| UA | 84±6 | 86±8 | 60±10 | 77±13 | 85±11 | 88±10 | 73±14 | 87±11 |
| C | 7±3 | 4±2 | 19±11 | 6±4 | 3±1 | 1±1 | 8±5 | 2±1 |

In the text, we also give more detail in order to explain how we based our temperature considerations on the works of C. Elvidge and Y. Liu:

The temperature value used in the selection process of a discrimination strategy is based on Elvidge et al. (2016) and on the recent work by Liu et al. (2018), who derived gas flaring temperatures of 1000 K to 2600 K from the VIIRS Nightfire database, depending on the type of operation (shale oil or gas, offshore, onshore or refinery). Most of the gas flares display temperatures between 1650 K and 1850 K. However, temperatures can occasionally be as low as 1300 K. We therefore test for the minimum andor for the maximum temperature for all the high-accuracy detections within a grid cell ($T_{min}$ and $T_{max}$, respectively). The temperature range reported by Elvidge et al. (2016) and Liu et al. (2018) overlaps with particularly hot detections from the coal chemical industry and steel plants. Therefore, additional criteria are needed for identifying gas flares in the hot source dataset.

In order to select the discriminating strategy we test several subsets of the gridded high-accuracy hot spot database. For each of the 8 subsets described in Table 1, a sample of 100 random onshore grid cells complying to the defined thresholds have been tested by examining high-resolution imagery (Google Earth) and the locations are classified into four categories:

**2.3 Determination of flared volumes and black carbon emissions**

- Please, explain the terms BCMmin, BCMmax, BCMbest in this order, to facilitate the comprehension.

This was updated as suggested.

- The emissions of black carbon (BC) from gas flares are estimated using reported emissions factors (EF). It could be useful to specify the formulation applied for their computation.

We have somewhat rearranged this paragraph and included a short introductory text to explain our approach:

The emissions of black carbon (BC) from gas flares are estimated using reported emissions factors (EF). Klimont et al. (2017) recognized the limited number of measurements of flaring emissions. Here, we attempt to consider the limited information available on the EF and maximize the use of the available information on the flare characteristics.

Schwarz et al. (2015) and Weyant et al. (2016) conducted field experiments in the Bakken formation (USA) and derived EFs of $0.57\pm0.14\ g.m^{-3}$ and $0.13\pm0.36\ g.m^{-3}$ (using the Single Particle Soot Photometer) or $0.28\ g.m^{-3}$ (using the Particle Soot Absorption Photometer), respectively. However, flared gas has not the same composition everywhere and Huang and Fu (2016) considered the regional variability of the EF. The authors applied the function which relates EF to the volumetric gas heating value derived in the laboratory by McEwen and Johnson (2012) to globally compiled gas composition data. Klimont et al. (2017) considered, for the Greenhouse Gas – Air Pollution Interactions and Synergies (GAINS) model, the EF derived by Schwarz et al. (2015) of $0.57\ g.m^{-3}$ for well-operated flares (i.e. Organisation for Economic Co-operation and Development (OECD) countries) and a maximum of $1.75\ g.m^{-3}$ for other countries. Stohl et al. (2013) used an EF of $1.6\ g.m^{-3}$ from a previous GAINS version. In the present work, we apply the same concept of a varying EF but use the flare temperature as an indication of the combustion completeness, instead of the country of origin as an indication of the flare operation. Flaring temperatures close to the adiabatic flame temperature for natural gas (around 2500 K) are associated with more complete combustion and therefore lower BC emissions. On the other hand, low flaring temperatures (700 K and below) are associated with higher BC emissions. Between the two extremes, the BC emission is scaled linearly as a function of the flaring temperature (see Figure 3). To the best of our knowledge, this is the first time that operating practices are taken into consideration when assigning the EF.

- GAINS: please, extend the acronym.

The acronym is explained in the text.

- You define flaring site a site with a temperature above 1500K. Why do you compute the EFs for lower temperatures?

The flaring site is defined as a grid cell for which the count of high-accuracy hot spots is larger than 5 and the maximum temperature is larger than 1500K. Although the maximum retrieved temperature must be larger than 1500K, temperature for individual high-accuracy hot spots within the grid cell may be lower than 1500K.

- With this methodology we estimate a wide range of possible activity (BCM) and BC emissions (g/m³)

The recommendation was followed in the updated manuscript although the unit for the BC emissions was kept as mass (g).

- Can you better explain this sentence, please? I do not understand it: "We conservatively assume that this range of possibilities represents 6 × σ, and report the uncertainty of the best estimates as 1 × σ".

For clarity, we have removed this part from the paper and report the best estimate together with the range.

**3. Results**
**3.1 Hot spots and flaring sites**
I have concerns about this section. Your paper focuses on gas flaring, the previous one (Caseiro et al., 2018) on hotspots. For this reason, you can join Figures 4, 5, 6 using three colors for discriminating hotspots, high confidence hotspots and flaring sites. Besides, I do not understand why you compare the SLSTR global detections for 2017 with the VNF in 2012. The VNF data for 2017 are available; you indeed use them in section 3.3.

Figures 4, 5 and 6 were merged into a single figure. Please see the resulting figure below.

[Figure]

- Russia (985) and the United States (917) are the countries with the highest number of flaring locations (Figure 7).
- The time series of the cumulative number of the high accuracy observations for the most active flaring location (in Venezuela, see Section 3.4) is shown in Figure 8. It shows flaring activity throughout the year. In my opinion, it is not useful and interesting. Remove Figure 8.

These suggestions were followed.

I think 3.2 and 3.3 are subsections of 3.1: they become 3.1.1 and 3.1.2.

We feel that the three sections bring enough information individually to be treated as being at the same level: 3.1 deals with the detection itself, 3.2 with their characteristics and in 3.3 we derive the activity. To make this clear, the title of 3.1 has been updated: "Flaring locations".

- Figures 10, 11 and 12 are not useful, in my opinion they could be removed. You can indeed add before Figure 9 and Figure 13, respectively, a global map (in color scale) showing the temperatures and RP values for the 6232 sites.

We have removed Figures 10, 11 and 12.

We have added figures for the global average T distribution:

[Figure]

**Figure 7.** Average flaring temperature (K) at the 6232 flaring locations.

and similarly for RP:

[Figure]

**Figure 9.** Average radiative power (MW) at the 6232 flaring locations.

- Figure 9. Distribution of the average retrieved  temperature (K) for the flaring locations

The suggestion was followed.

- The average temperature at the flaring locations approximately ranges from 950 K to 2250 K. This is slightly lower than the range reported by Liu et al. (2018) (please, can you specify the values) who used VIIRS Nightfire data, as expected from our previous study (Caseiro et al., 2018). It confirms the bi-modal distribution with modes around 1750 K and 1200 K that is has also been observed by VIIRS.

The range given in Liu et al. (2018) was specified in the manuscript.

- The section "Comparison with VIIRS Nightfire" should be modified. As before explained, being the focus of your work the gas flaring, you should compare your results with the VNF flaring sites (available at https://www.ngdc.noaa.gov/eog/viirs/download_global_flare.html), avoiding to select these sites among the VNF combustion sources applying the criteria used for SLSTR.

The section was rewritten taking into consideration your suggestions. The section is now at the end of the "Results" chapter and we included a comparison of the activity (flared volumes) as well.

- You never cite Figure 14 in the paper. The figure is not useful, as figures 10-12.

Figure 14 was removed from the manuscript.

**3.4 Flared volumes** (new 3.2)
As before, you should use BCM data available at https://www.ngdc.noaa.gov/eog/viirs/download_global_flare.html for the comparison with your estimates in 2017. It would be interesting the map of the global distribution of BCMbest. In Figure 20 you could add the distribution derived by the VNF data elaboration.

**3.5 BC emissions** (new 3.3)
As for BCM, you can add a global map of BC emissions.

This section was reworked also following the recommendations from the other reviewer and the short comments. It now includes VNF data from 2017 as suggested. Please see the updated manuscript.

> **Conclusions**
> To reorganize based on new suggested analyses. In any case:

This section was reworked also following the recommendations from the other reviewer and the short comments. Please see the updated manuscript.

> - The sentence "We present a new gas flaring discrimination procedure, based on two characteristics of gas flares: persistence and temperature" is not correct. This procedure is not new, being the one most used to identify gas flares. Respect to your methodology, you simply add a temperature filtering to improve the detection of flaring sites.

We have updated this sentence of the conclusions: "We adapt the procedure most commonly used to discriminate gas flares (based on two characteristics of gas flares: persistence and temperature) to our specific hotspot detection methodology."

> - "Additionally to the detection we present a way to assess the volume of flared gas": is not true. You apply a widely declared model developed by Elvidge et al (2016) to compute monthly flared volumes, adding a scaling factor, which takes into account the operation time of the sites.

We have reworded the first two sentences of this paragraph: "Additionally to the detection we assess the volume of flared gas based on the observed relationship between the flared volume and observed flare radiative energy."

---

## Author Comment (AC2) · 29 Apr 2020

Dear Chris Elvidge, thank you very much for your valuable and constructive comments. We have followed most of your suggestions and we find that our paper is now generally improved.

In this document, your original comments are framed by a box and our answer follows.

> Review of "Gas flaring activity and black carbon emissions in 2017 derived from Sentinel-3A SLSTR" The paper attempts to locate all the active gas flares of 2017 and estimate their flare gas volumes using nighttime data collected by the Sea and Land Surface Temperature Radiometer (SLSTR) instrument flown on-board the Copernicus satellite Sentinel-3A. The basic detection algorithm for the individual nights of data follows the VIIRS nightfire (VNF) method and appears to be solid. But the steps used to go from the individual nights of data to the annual summary are questionable and should be revisited.

> 1. The SLSTR results found 6232 flaring sites in 2017. This compares to over 10,000 flares reported by the VIIRS nightfire team for 2017 (https://eogdata.mines.edu/download_global_flare.html).

The section on the comparison with VIIRS Nightfire has been totally rewritten and is now at the end of the Results chapter. We now compare our results to the gridded detections and activity data for 2017 provided by the VIIRS Nightfire team.

We argue that both methodologies globally agree, in that they capture roughly the same flaring regions around the world. However, a closer look reveals that those regions are more populated by the VIIRS Nightfire results than by our own. We trace back that behaviour to small geolocation inaccuracies, the clustering of hot pixels in our algorithm (against the analysis of local maxima) and the VIIRS larger swath which provides more opportunities to detect a flaring site. We attempt to quantify the contribution of these factors to the observed difference in terms of flaring locations between both methodologies. Of the 10185 flaring locations produced by the gridding of the VNF data, 2964 are coincident with the flaring locations presented here, 1507 are adjacent and 1651 are less than 4 grid cells away, which can be interpreted as due to geolocation inaccuracies and the clustering of hot pixels. Finally, 4063 are "distant", i.e. more than 4 grid cells away from the closest SLSTR flaring location. A closer analysis reveals that our methodology also captures activity at those locations, though not enough to be classified as flaring following our criteria. This indicates a low activity and, indeed, the associated BCM in the VNF record is low (8.67 BCM, a few percent of the global total of 151 BCM).

In summary, we attribute different flaring locations in the datasets as follows:

- Adjacent (7.3 BCM in VNF): a mix of geolocation error and clustering
- A distance from adjacent up to 4 grid cells (3.1 BCM in VNF): a mix of clustering and intermittent operation
- Larger distances (8.7 BCM in VNF): intermittent operation and more detection opportunities by VIIRS due to larger swath.

> 2. Many flares are intermittent. The nightly flare detection data does not contain sufficient information to account cloud and solar contamination effects that could effect the annual flared gas volume calculation. Hence the annual characterization of flared gas volume should calculate the "duty cycle" or "percent frequency of detection" for each flaring site. The VNF team makes the calculation based on flare detection numbers in the set of nighttime cloud-free observations made of the site during the year. Because the nightly VNF product only contains the detections – the annual analysis includes an

inventory of the cloud state (cloudy or clear) for the nights lacking VNF collection that are free of solar contamination. The VNF method excludes both sunlit and cloudy observations in the calculation of flaring site duty cycle. The method reported in this paper (section 2.3 and Figure 2) is woefully inadequate and appears to have resulted in a drastic underestimation of annual flared gas volume in Russia. I suspect that the method in Figure 2 does not account for solar contamination outages during summer months – as shown below with VNF for a flare in northern Siberia.

[Figure]

**Fig. 1.** Temporal profile VNF SWIR radiance for a flare in Siberia. Each year there is an outage period due to solar contamination.

We acknowledge that the method for correcting for variable observation opportunity is less sophisticated than the one employed in VNF. It should, however, be capable of partly correcting the effect partial and complete solar contamination. The underlying assumption is that SLSTR happened to observe a few gas flaring locations in mostly cloud-free condition during 2017. With almost continuous operation, these should be the locations with the largest numbers of detections. In this case, the numbers of detections are also estimates for the numbers of detection opportunities, including the effects of number of satellite overpasses and solar contamination. The number of satellite overpass times increases for higher latitudes; this should increase this largest number of detections for higher latitudes. On the other hand, the summer period during which more observations are contaminated with solar radiation is longer for higher latitudes. This has the opposite effect of reducing the largest number of detections for higher latitude. The latter effect appears to be stronger, given the behaviour depicted in Fig. 2. An additional error occurs due to the variability in average percentage of cloud cover, which the presented method approximates to be constant zonally and to vary smoothly meridionally. Therefore, the method is considered to be applicable with less effort but also less accuracy than a detailed recording of the detection opportunities for each flaring site. In particular, this method should improve the results when only information on hot spot detections but not on detection opportunity is available.

> 3. The paper lacks detail on the method used to discriminate clear versus cloudy observations. In addition, the paper makes several assertions that should be rechecked.

The discrimination of clear versus cloudy observations is performed based on the cloud mask in the SLSTR L1b product during the hot spot detection procedure, which is described in our earlier paper and simply applied in this one. We agree that a second processing of the SLSTR L1b products to calculate the detection opportunities for each flaring site based on its individual solar and cloud contamination would be a more accurate process. However, due to budget (the project has been over for more than 2 years now) and time (the main authors of the study are now in new positions) constraints, it is not possible for the authors to conduct this now. It should certainly be considered for possible further developments of the methodology.

4. The paper makes several claims that the "SLSTR-based methodology is able to detect smaller gas flares". No evidence is presented to back up this claim.

The claim was based on the night-time availability of a second SWIR channel. The statement was completed in the manuscript (see the answer to the next comment).

5. The paper state that the VNF product only uses a single shortwave infrared channel. This was the case for early VNF data. However, from January 2018 forward VNF from two satellites has included two SWIR bands.

We have updated the information in our manuscript (section 2.1 Hot spot detection and characterization): [referring to the comparison conducted in our previous paper] The results showed a good agreement of our hot source detection when investigating persistent hot spots with the advantage of the Sentinel-3A's SLSTR algorithm in detecting and quantifying smaller flares, due to the night-time availability of a second SWIR channel. Although this was the case at the time of writing our previous paper, from January 2018 VIIRS Nightfire uses two SWIR channels at night and the detections are conducted by two VIIRS instruments.

The reference to the outdated feature was removed from the end of the introduction.

My recommendation is that the paper undergo major revision and a second round of peer review prior to publication. The authors should make a specific comparison against the VNF product from 2017 to better understand difference between the SLSTR and VIIRS flaring sites and flared gas volumes. Are there specific geographic regions where one system detects more flares or more flared gas volumes? Since the instruments and detection algorithms are so similar, the authors should figure out the reason behind the discrepancies. To make a direct comparison of the combustion source detection limits with VNF, the authors can follow the methods outlined in https://www.mdpi.com/2072-4292/11/4/395

This crosschecking with VNF could lead to major improvements in the gas flaring results from SLSTR and a far better paper.

We have considerably revised our paper and have added several new analysis and figures to better illustrate the results of our work. In particular, we have added a comparison the VNF product from 2017 as recommended. The suggested further analysis along the lines of the paper recommended by the reviewer would certainly be worthwhile but is considered to be somewhat out of the scope of the present submission. In the Elvidge *et al.* 2019 paper, the authors analyze the hotspots retrieved using not only a SWIR channel as primary detection, but also a MWIR channel. In our present work, we focus on the detections achieved using the SWIR as primary channel.

---

## Author Comment (AC3) · 29 Apr 2020

Dear Daniel Fisher,

Thank you very much for your valuable comments. We have followed your suggestions and we find that our paper is now improved in general and more thoroughly compared with the most recent literature on the subject.

In this document, your original comments are framed by a box and our answer follows.

In this paper the thermal anomaly detection and characterisation algorithm developed in Casiero et al. (2018), based on NightFire, is applied to Sentinel-3 SLSTR data with the objective of evaluating global gas flaring radiant heat output and the associated estimates of black carbon emissions. The resulting thermal anomaly outputs, generated for 2017, are subjected to various filters to partition flaring and non-flaring anomalies, with the most crucial step being the application of a persistency test of more than five observations in the 12 month evaluation period in addition to a minimum temperature limit of 1500 K, which must be exceeded. A normalisation process is then applied to the detected gas flaring sites to account for differences in sampling opportunities to generate an adjusted measure of radiant heat output for each flaring site. Black carbon emissions estimates are then generated from the adjusted radiant heat estimates using appropriate emission factors. Finally comparison against various other datasets are made to assess the validity of the SLSTR generated datasets.

Whilst in general the paper is reasonably well written, there is a potentially significant flaw in the gas flare characterisation approach which must be investigated and addressed before publication can be recommended. In Elvidge et al. (2016) and Fisher et al. (2019) it is demonstrated that a substantial proportion of global gas flaring radiant heat output arises from a small subset of flaring sites global (50% of output comes from 5-10% of all flares). Some of these flares are extremely radiant such as the Punta de Mata site in Venezuela identified by the authors (and elsewhere), and they attribute 0.623 BCM of flared gas to this site in 2017. In comparison, Elvidge et al (2016) identified that 1.13 BCM of gas was flared at this site in 2012. The discrepancy between these two values is concerning, particularly as Venezuela has been shown to have had a very large increase in gas flare radiant heat output since 2012 (Fisher et al., 2019), and given the characteristics of gas flaring (e.g. most radiant heat being produced by a small subset of sites) one would expect that the most active flaring site in Venezuela to have show at least some increase, and not the reported decrease.

I think that this result may be arising from a potential issue with the channels used by the algorithm. In Elvidge et al. (2019) the issue of saturation in the (M11) 2.2 m and (M12) 3.7 m VIIRS channels is identified and I would expect that a similar issue is occurring with SLSTR, particularly given the enhancement in pixel resolution from 750 m2 to 500 m2. The channels employed in the Casiero et al. (2018) for the estimation of gas flare radiant heat output are S5 (1.6  $\mu$ m), S6 (2.2  $\mu$ m) and S7/F1 (3.7  $\mu$ m). Given its specifications, the S7 channel likely saturates on a regular basis over gas flares, and this is identified, as is the potential for using the F1 channel when this occurs. However, no assessment of the

saturation characteristics of the SWIR channels is made, and whilst the S5 channel likely does not saturate, the same cannot be said for the S6 channel. Furthermore, being a relatively new instrument the performance of the F1 channel has not been evaluated, and it needs to be demonstrated that it reaches the specified dynamic range performance levels (the same can be said for the SWIR channels). If any of the channels are saturating then the effects on the retrieved radiant heat can be significant as shown in Elvidge et al. (2019), and in the current configuration of the algorithm used in this paper these saturation events may well be being missed. I expect, given that a significant proportion of radiant heat output arises from a subset of highly radiant flaring sites, the impact of saturation on the reported global total of radiant heat output and in turn BCM could well be significant and must be explored.

To do so I would recommend:

(1) evaluating the Planck curves of the most radiant flare sites identified globally and check the deviations of the various spectral observations from the curves that might indicate saturation of specific channels, giving at least an indication of saturation effects on SLSTR.

We already address the reviewer's concern of S7 and S9 saturation by using F1 and F2 when necessary (see also Caseiro et al. 2018). Saturation of the SWIR channels do not lead to a good fit and do not converge. Please find all the Planck curve fits within the most active flaring location at the end of this document.

(2) I would recommend applying the single channel radiant heat estimation approach of Fisher and Wooster (2018) developed in part for application to the S5 channel of the SLSTR sensor and see if any significant differences are observed, this would be a very straightforward comparison.

We added a new subsection where we apply the single SWIR method (Fisher and Wooster 2018 and 2019) to our detections and compare the results in terms of radiative power (RP), see figure below. We also compare our results with the results from Fisher and Wooster 2019. This shows that our method is biased high compared to the SWIR method and also gives larger maximum FRP values. Therefore, the FRP calculation should not lead to a low bias for the largest gas flares as suspected by the reviewer.

(3) I would then suggest that if a discrepancy for these larger flaring sites is found that the S3 channel is included as an additional constraint for these very large flares to try and improve the radiant heat estimates.

The NIR channels S3 (0.865  $\mu$ m) and/or S4 (1.375  $\mu$ m) are strong candidates to constrain more precisely the dual Planck curve fit. Both these channels are turned off at night, but during the commissioning phase of the satellite, which corresponded to the development phase of our methodology, they were switched on at night by ESA for a limited time period. Our conclusions were that the signal-to-noise ratio was too low for extracting additional information.

(4) It would be likely be useful to compare also against flare counts from Fig. 7 in Fisher and Wooster (2019) in addition to the NightFire comparison.

This has been added to the manuscript, see answer to recommendation #2.

Lastly, some key references are missing from the paper and should be included:

Anejionu, O.C., 2019. Rationale, historical developments and advances in remote sensing of gas flares. International Journal of Remote Sensing, 40(17), pp.6700-6719.

Elvidge, C.D., Zhizhin, M., Baugh, K., Hsu, F.C. and Ghosh, T., 2019. Extending nighttime combustion source detection limits with short wavelength VIIRS data. Remote Sensing, 11(4), p.395.

Fisher, D. and Wooster, M., 2018. Shortwave IR Adaption of the Mid-Infrared Radiance Method of Fire Radiative Power (FRP) Retrieval for Assessing Industrial Gas Flaring Output. Remote Sensing, 10(2), p.305.

Fisher, D. and Wooster, M.J., 2019. Multi-decade global gas flaring change inventoried using the ATSR-1, ATSR-2, AATSR and SLSTR data records. Remote Sensing of Environment, 232, p.111298.

1 + 2 + 4 were not originally included because they came out after or towards the end of the time of writing. All four are now cited in the paper.

Other references referred to here:

Caseiro, A., Rücker, G., Tiemann, J., Leimbach, D., Lorenz, E., Frauenberger, O. and Kaiser, J., 2018. Persistent Hot Spot Detection and Characterisation Using SLSTR. Remote Sensing, 10(7), p.1118.

Elvidge, C. D., Zhizhin, M., Baugh, K., Hsu, F.-C., and Ghosh, T.: Methods for Global Survey of Natural Gas Flaring from Visible Infrared Imaging Radiometer Suite Data, Energies, 9, 14,

---

## Author Comment (AC4) · 29 Apr 2020

Dear reviewer,

In this document, your original comments are framed by a box and our answer follows.

1. The methodology section is too brief and the reader needs to go through many other publications to gain even an overview of the current one. This is inconvenient for the general readers. E.g. in line 12 of page 3, instead of directly pointing to Caserio et al. (2018), just very briefly give an overview of the salient features of the algorithm.

Similarly, for line 19 of page 3, explain why analyzing he cluster of contiguous hot pixels advantageous than the spatial maxima method. Additionally, what could be the disadvantages of this method and how are they taken care of: e.g. What are the possibilities that pixels representing different intensities taken as an average may lead to an overall over or underestimation for a grid or lead to a mixing of two very different signals.

We have added several sentences in the beginning of the methodology section. However, since the methodology has already been published and this paper is intended to describe a dataset, we preferred to keep comments on the methodology short in order to keep the focus of the reader on the data.

2. Section 2.2.2 : state the background, advantages and disadvantages for using the Equations 1 and 2.

Equation 1, user's accuracy: The User's Accuracy is the accuracy from the point of view of a map user. This metrics represents the frequency with which a class on the map corresponds to the ground truth. In our case, the User's accuracy is quantified between at least the verified flaring locations and at most the sum of the verified and the likely flaring locations.

Equation 2, commission error: Commission errors are calculated by reviewing the classified sites for incorrect classifications. In the present work, the commission error is quantified between at least the locations without any industry or infrastructure and at most the sum of the locations without any industry or infrastructure and the unlikely flaring locations.

We have updated the manuscript accordingly.

3. Section 2.3: The BC estimation formula seems too oversimplified. E.g. equation 4 does not take into account the different flares leading to different amounts of BC emissions over the year. This may be a reason leading to underestimation w.r.t. other inventories: I believe authors should instead try something like a weighted average or a more representative method to arrive at a better estimate.

The BC emission factor takes into account the different state of a flare in terms of burning efficiency in that it is determined as a function of temperature (please refer to Figure 3). While appearing to be simple, we argue that this methodology goes beyond differentiating between static flare types by parameterising the underlying physical reason (flare temperature resp. burning efficiency). It is therefore able to account for different and varying emission factors. Equation 4 is used to compute the total activity or emission of a flaring site. We agree that the estimation of the number days of operation for each flaring site is the part of the entire method with the most assumptions and thus

the largest potential source of errors. However, we consider it to be a valid approach when only detections of hot spots are available as input. Please refer to our answer to the corresponding comments by Chris Elvidge for more details.

4. The diagrams need to be more complete by themselves e.g. y label missing in fig 11, 12 (only writing it within caption is not sufficient), cbar label missing in fig 10

Figure 10, 11 and 12 were removed from the manuscript following the recommendation by the other anonymous reviewer.

Page 2 Line 11: Contributions to what e.g. CO2 equivalent?

Page 2 Line 14: Replace 'emitted' by 'from'

Page 3 Line 4: Replace 'chapter' by 'section'

Page 21 line 9: what is Upstram?

Page 24 Line 4: Replace 'allow' by 'allows'

The Upstream oil and gas industry is that part of the oil and gas industry which includes searching for potential underground or underwater crude oil and natural gas fields, drilling exploratory wells, and subsequently drilling and operating the wells that recover and bring the crude oil or raw natural gas to the surface. An addition was made to the text.

All the other recommendations were followed.

---

## Referee Report (RR1)

[referee-annotated manuscript omitted]

---

## Author Response (AR3)

**Topical Editor Decision: Publish subject to technical corrections** (18 Jul 2020) by Vinayak Sinha
Comments to the Author:
Dear Authors,
Thank you for the revisions and addressing the reviewers' concerns.
The version of the MS submitted by you still has some editing and highlighting remarks in red and yellow.
Please take care to remove these and resubmit a final clean version without any edits marked in, so that the paper can be published in ESSD.
Thank you for your patience duirng the COVID delays and also considering ESSD for your work.
All the best-
Vinayak

Dear Vinayak,

Thank you for your answer. We update a clean final version without annotations.

Kind regards

Alexandre Caseiro and co-authors